# Supply-side and demand-side factors influencing uptake of malaria testing services in the community: lessons for scale-up from a post-hoc analysis of a cluster randomised, community-based trial in western Kenya

Joseph Kirui [1], Josephine Malinga,[2] Edna Sang,[3] George Ambani,[1] Lucy Abel,[1] Erick Nalianya,[3] Jane Namae,[4] Matthew Boyce,[5] Jeremiah Laktabai,[1,4] Diana Menya,[6] Wendy O'Meara [2,6]

For numbered affiliations see end of article.

**Correspondence to**
Dr Joseph Kirui;
josepheddykipkoech@gmail.com

## ABSTRACT

**Objectives** Maximising the impact of community-based programmes requires understanding how supply of, and demand for, the intervention interact at the point of delivery.

**Design** Post-hoc analysis from a large-scale community health worker (CHW) study designed to increase the uptake of malaria diagnostic testing.

**Setting** Respondents were identified during a household survey in western Kenya between July 2016 and April 2017.

**Participants** Household members with fever in the last 4 weeks were interviewed at 12 and 18 months post-implementation. We collected monthly testing data from 244 participating CHWs and conducted semistructured interviews with a random sample of 70 CHWs.

**Primary and secondary outcome measures** The primary outcome measure was diagnostic testing before treatment for a recent fever. The secondary outcomes were receiving a test from a CHW and tests done per month by each CHW.

**Results** 55% (n=948 of 1738) reported having a malaria diagnostic test for their recent illness, of which 38.4% were tested by a CHW. Being aware of a local CHW (adjusted OR=1.50, 95% CI: 1.10 to 2.04) and belonging to the wealthiest households (vs least wealthy) were associated with higher testing (adjusted OR=1.53, 95% CI: 1.14 to 2.06). Wealthier households were *less* likely to receive their test from a CHW compared with poorer households (adjusted OR=0.32, 95% CI: 0.17 to 0.62). Confidence in artemether–lumefantrine to cure malaria (adjusted OR=2.75, 95% CI: 1.54 to 4.92) and perceived accuracy of a malaria rapid diagnostic test (adjusted OR=2.43, 95% CI: 1.12 to 5.27) were positively associated with testing by a CHW. Specific CHW attributes were associated with performing a higher monthly number of tests including formal employment, serving more than 50 households (vs <50) and serving areas with a higher test positivity. On demand side, confidence of the respondent in a test performed by a CHW was strongly associated with seeking a test from a CHW.

## STRENGTHS AND LIMITATIONS OF THIS STUDY

⇒ This analysis draws on a large multidimensional dataset describing uptake of community-based services (intervention recipients) and delivery of services (intervention providers) in the context of a cluster randomised implementation trial, thereby allowing us to examine the strengths and shortcomings of the intervention by triangulating data from two unique perspectives.

⇒ These data were from a representative population sample which includes respondents who were successfully reached with the intervention as well as those who were not tested for malaria before taking antimalarials and may require alternative approaches in order to achieve adequate coverage of diagnostic testing before treatment with antimalarials.

⇒ As a secondary analysis of trial data, we were limited in the scope of the data available and not able to fully explore the interaction between the community health workers and households nor the complex determinants of health-seeking behaviour.

**Conclusion** Scale-up of community-based malaria testing is feasible and effective in increasing uptake among the poorest households. To maximise impact, it is important to recognise factors that may restrict delivery and demand for such services.

**Trial registration number** NCT02461628; Post-results.

## INTRODUCTION

Despite decades of control efforts, malaria remains a major public health problem. In 2019, an estimated 229 million cases of malaria occurred worldwide, most of which were in the WHO African Region (215 million or 94%).[1] In the same year, there were an estimated 409 000 deaths from malaria globally,

with Africa again accounting for 94%.[1] Children under 5 years are the most vulnerable group affected by malaria and in 2019, 67% of malaria deaths were in this group.[1]

Prompt malaria diagnosis either by microscopy or rapid diagnostic tests (mRDTs) is recommended by WHO for all patients with suspected malaria before they are given treatment.[2] There are already 95 countries that have adopted the policy of testing all patients with suspected malaria before being treated, and this number is increasing. Diagnostic testing is free of charge in the public sector of almost 90 countries.[3] Early and accurate diagnosis is essential both for effective management of the disease and for strong malaria surveillance. Parasite-based diagnostic testing significantly reduces illness and death by enabling health providers to swiftly distinguish between malarial and non-malarial fevers and select the most appropriate treatment.[4]

The Kenya National Control Program adopted the Test and Treat policy in 2010 and this has been cascaded to the community level. The mRDTs are recommended for parasitological diagnosis in public health facilities where microscopy is not available and through community health workers (CHWs) in some regions. Despite these efforts, most people in rural Kenya still opt for self-treatment and seek care in the retail health sector.[5 6] In our study setting, mRDTs are not available in pharmacies and chemists; therefore, patients who purchase antimalarials in the retail sector receive no parasitological diagnosis before treatment. Studies have shown that targeting of antimalarials to true malaria cases in the retail sector is very low: as few as 20% of people purchasing antimalarials are suffering from malaria[7 8] and as high as 70% of people with malaria do not purchase an artemisinin combination therapy.[9]

The deployment of mRDTs through trained CHWs could improve access to diagnostic testing for populations with limited access to health facilities. In a large, cluster randomised trial,[6] we evaluated an intervention that targeted individuals who seek care in the retail sector by offering free community-based malaria diagnostic testing through CHWs. Following testing, those with a positive test result received a voucher to obtain first-line antimalarials at a nearby pharmacy at a heavily subsidised price. The intervention demonstrated a significant improvement in testing before treatment.

Ideally, there should be adequate demand for the commodity or opportunity, and supply should be able to meet demand. In our cluster randomised trial, despite improvements in malaria testing before treatment of fever, we still observed that only 55% of those with recent illness reported having a malaria test before taking drugs and 21% still purchased antimalarials over the counter without a test,[6] representing substantial scope for further improving the reach and impact of this intervention. We predicted that the intervention could have led to higher testing rates and sought to understand whether the impact may have been limited by either demand for, or supply of, testing or both.

Here we report a secondary analysis of trial data collected at 12 and 18 months after the initiation of our cluster randomised trial of community-based diagnostic testing in western Kenya.[6] We sought to understand what factors might increase or decrease demand for testing from the community, compared with factors that may have limited supply of testing by CHWs. A comparison of constraints on the demand side and supply side could help us understand the conditions under which the intervention operated well or was hindered by a mismatch in supply and demand. The results will be instrumental for improving the design of community-based interventions delivered at scale.

## METHODS
### Description of the main trial
This is a post-hoc analysis of data collected within intervention clusters during a large community-based cluster randomised trial. The main results of the trial have been reported elsewhere.[6] The purpose of the parent trial was to encourage clients to receive testing for malaria before seeking treatment. This was accomplished by training and equipping CHWs to perform mRDTs for malaria and to provide clients who test positive for malaria with a voucher that allows them access to subsidised high-quality antimalarial drugs through participating pharmacies.[6] The trial was conducted in 32 community clusters in western Kenya (population approximately 160 000). Eligible clusters had retail outlets selling artemether–lumefantrine (AL) and existing CHW programmes and were randomly assigned 1:1 to control and intervention arms. In intervention areas (comprised of 16 community units), CHWs were available in their villages to perform mRDTs on demand for any individual >1 year of age experiencing a malaria-like illness. The mRDT-positive individuals received a voucher for a discount on a quality-assured AL, redeemable at a participating retail medicine outlet. In control cluster (composed of 16 community units), CHWs offered a standard package of health education, prevention and referral services. We conducted four population-based surveys—at baseline, 6 months, 12 months and 18 months—of a random sample of households with fever in the last 4 weeks to evaluate predefined, individual-level trial outcomes.

In Kenya, CHWs work on a volunteer basis and there are no formal education requirements. The ability to read and write are criteria for selection by the Ministry of Health (MOH). The role of CHWs as defined by the MOH is to promote good health through providing health education, basic treatments (such as first-aid) and referrals to healthcare facilities. The CHWs report to community health extension workers who are employees of MOH and are trained community health nurses mandated to supervise the activities of CHWs in the community units (CUs). In intervention areas, existing CHWs were trained to carry out the intervention. Two hundred seventy-two

CHWs participated in offering testing, a median of 18 CHWs per cluster (range: 10–24).

CareStart HRP2-based malaria mRDTs were used for testing. CareStart HRP2 detects *Plasmodium falciparum*, which is the main species of parasite responsible for the majority of malaria cases in Kenya.[10] In addition to mRDT, the study team provided CHWs with the necessary supplies to conduct the mRDT. CHWs were free to perform tests in the location(s) of their choice (eg, in their own homes, at clients' homes, etc), as long as the location allowed for proper and sufficiently private performance of the procedure.

The CHWs tested febrile patients at community level and issued AL vouchers to patients with positive results. This CHW model differs from other settings implementing community case management of malaria where the CHW is permitted to both perform the mRDT and provide antimalarials directly to the client. The voucher redemption period was 3 days whereby after that, the medicine shops were not allowed to collect the voucher. Patients with positive malaria results bought AL from medicine shops by redeeming their vouchers and leaving the voucher at the shop. The study team collected the redeemed vouchers from the medicine shops on a biweekly basis and reimbursed the difference in costs.

## Community surveys

A community-based survey was conducted to collect individual-level study outcomes based on a population-based survey sampling strategy. This was a repeated cross-sectional household survey at four time points: baseline (pre-intervention), 6 months, 12 months and 18 months post-baseline. The survey was conducted in all the 32 CUs, 16 intervention and 16 control. Households were randomly selected by systematic random sampling and one person with fever in the last 4 weeks was interviewed in any household. If more than one eligible fever was found in a household, one participant would be selected based on alphabetical order of the given names. Details of community surveys' sample size calculation are described elsewhere.[11] Briefly, the target sample size of eligible individuals with a fever in the last 4 weeks was 640 per arm (ie, 40 per CU) at each time point, which—assuming 22.2% of households would have an eligible member—would require contacting 5766 households. The starting point (index house) and sampling interval were different in each survey round in order to minimise repeat surveys across multiple rounds. We restrict our analysis here to the 12-month and 18-month surveys in order to evaluate the effects of interest in the time period when the intervention effect could be measured.[6]

Basic demographic information about the participants and the households was collected. Socioeconomic scores were calculated from an asset index and then grouped into quintiles as described previously. Euclidean distance from a household to the nearest health facility was calculated from Global Positioning System coordinates. In the final survey, participants' beliefs about their illness,

malaria risk, diagnostic testing and treatment were elicited by asking them about the severity of their illness, their confidence in positive test result and a negative test result, and how certain they were that they would recover from malaria after treatment with AL. 'Perceived prevalence' was defined as the proportion of fevers that the respondent believed would be due to malaria among 10 people with fever from their village. High perceived prevalence was assigned to those who responded 8–10 out of 10, medium prevalence to those who responded 4–7 out of 10 and low to those who responded 0–3 out of 10. The survey tool can be found in online supplemental materials.

## Midpoint CHW survey

As part of the process evaluation, at the midpoint of this study, interviews were conducted with a random sample of 70 CHWs (median of 4 per CU) who were trained to conduct mRDTs (intervention CUs). The primary objective of these interviews was to determine the CHWs' satisfaction with their role in the malaria project and the feasibility of their continued involvement. Secondary objectives included eliciting the main challenges the CHWs have faced in implementing the study, determining unanticipated burdens on the CHWs' other activities (including their responsibilities as CHWs outside of the study as well as their non-CHW responsibilities) and identifying changes in the types of services they provide and the recipients of their services.[12]

Likert scale questions were also asked, in which the interviewers read a statement aloud and the CHWs were asked to respond how much they agreed with the statement on a 5-point scale ('strongly agree', 'somewhat agree', 'neutral', 'somewhat disagree', 'strongly disagree'). For this analysis, several variables were derived from the survey data. The client score was constructed as a composite score based on four Likert scale questions—whether CHWs agreed that: clients trust the results of a positive mRDT by the CHW, they trust a negative mRDT, they follow the CHW's advice if the test was positive, they follow advice if the test was negative. A response of strongly agree=2, somewhat agree=1 and neutral/somewhat disagree/strongly disagree=0. If the sum of these four responses was 6 or greater, the score was 'high'. If less than 6, the score was 'low'. The survey tool can be found in online supplemental materials.

These same CHWs also performed an mRDT under observation of the study supervision team and were scored on each step using a 20-step checklist that classified steps as relating to safety, accuracy and treatment. The checklist was developed by the study team based on previous research studies to guide training and support supervision. Those who completed at least 17 of 20 steps correctly had a 'high' score for mRDT performance and <17 had a 'low' score.

## Demand and supply

To investigate factors that might have influenced the success of the intervention, the analysis was divided into demand and supply factors. On the demand side, the analysis investigates factors that might influence the community member's outcome of having a test, first from any source and then from the CHW. Only respondents with complete data on the variables of interest were included. On the supply side, the analysis investigates the factors associated with the testing output (ie, supply) at the CHW level, measured as the monthly number of tests conducted by each CHW. These factors include: sociodemographic characteristics of the CHW, number of households under the CHW's care, years of experience of the CHW, the number of other activities the CHW engages in under their role as a CHW (under the assumption that an increasing number of activities may compete for their time, the CHW provided a list of activities and these were summed together), the perception of the intervention by the CHW, the skill of the CHW in conducting mRDTs and the overall prevalence of malaria in their community.

## Outcome measures

The primary outcome is the uptake of testing in the intervention clusters, defined as the proportion of fevers in the previous 4 weeks that received a diagnostic test from any source. The second primary outcome is the uptake of testing from a CHW among those who received a diagnostic test from any source.

The secondary outcome was the testing volume, defined as the mean monthly number of tests performed by each CHW. Monthly number of tests per CHW was available from programme data collected at monthly supervision visits.

## Statistical analysis

A mixed-effects logistic regression analysis was used to evaluate the primary outcome measures on the uptake of testing and investigate risk factors that might have contributed to the outcomes of the intervention reported previously.[6] We adopt a mixed-effects logistic regression to account for clustering at the level of the CU. The outcomes were evaluated only for the CUs in the intervention arm and summaries provided.

For the primary outcome measures to evaluate the uptake of testing overall and from a CHW, two logistic models were generated. The first model included all individuals who reported fever in the last 4 weeks (ie, all eligible participants enrolled in the survey). The second model includes a subset of those in the first model, specifically those who were tested at any source. For both models, we accounted for clustering within the CU in the univariate analysis and we accounted for clustering, and individual-level covariates like respondent's age, gender, level of education, occupation, socioeconomic status, geographical proximity to healthcare provider, the time they waited after falling ill before they sought a test,

the level of knowledge and opinions about malaria and malaria diagnosis in the multivariate analysis.

The logistic regression model is as shown below:

$$log\left(\frac{p_{ij}}{1-p_{ij}}\right) = \beta_0 + \beta_1 X_{1j} + \ldots + \beta_m X_{mj} + u_j$$

where $p_{ij}$ is the probability of the outcome for each individual $i$ in cluster $j$, $\beta_0$ is the model intercept, $\beta_{1-m}$ are the coefficients for each potential risk factor $X_{1-m}$ and $u_j$ represents the random effects for cluster $j$. The structures for both models are the same; however, for model 1, any test=1 and no test=0 and for model 2, tested at CHW=1 and tested elsewhere=0.

Fixed effects are measured using an OR and included the 95% CIs. Independent categorical variables were screened for multicollinearity with a Cramér's V statistic. In case two variables showed signs of correlation (V>0.7), the variable least associated with the response variable was omitted.[13]

For the secondary outcome measure of CHW testing, a mixed-effects linear regression model was fitted to account for clustering at the CU. This model investigates the outcome of mean RDTs performed each month by the specific CHW and includes potential CHW-level covariates like CHW age, marital status, education level, number of households for which they were responsible, previous training on malaria case management or mRDTs, years of experience as a CHW and formal employment status.

The linear regression model is as shown below:

$$Y_{ij} = \beta_0 + \beta_1 X_{1j} + \ldots + \beta_m X_{mj} + u_j + \varepsilon_{ij}$$

where $Y_{ij}$ is the outcome for each individual $i$ in cluster $j$, $\beta_0$ is the model intercept, $\beta_{1-m}$ are the coefficients for each potential risk factor $X_{1-m}$, $u_j$ represents the random effects for cluster $j$ and $\varepsilon_{ij}$ is the error term.

All statistical analyses were done in STATA V.15 (StataCorp).

## Patient and public involvement

None.

## RESULTS

A total of 272 CHWs in 16 intervention CUs were trained to perform mRDTs and 36 retail shops were enrolled to redeem vouchers. The intervention was launched on 21 July, 17 August and 22 September 2015 in Bokoli, Kiminini and Ndivisi subcounties, western Kenya, respectively, and continued until 5 May 2017.

## Demand-side factors for use of malaria diagnostic testing

In total, 3719 households were approached, 1877 had an eligible fever in the last 4 weeks, 56 did not know if there was an eligible fever, 0 refused and 1738 completed the survey with no missing answers (92.6%). The community-based demand analysis relies on 1738 complete surveys from participants in randomly selected households in intervention clusters with an acute illness in the last 4 weeks (table 1). Community members were interviewed

**Table 1** Participant characteristics at the 12-month and 18-month surveys and association between participant characteristics and malaria diagnostic testing

| | Sample characteristics | Univariate analysis | | Multivariate analysis | |
|---|---|---|---|---|---|
| | | Unadjusted OR (95% CI) | P value | Adjusted OR (95% CI) | P value |
| | (N=1738) | | | | |
| Outcome=tested by any source | 948/1738 (54.6%) | | | | |
| Time to seek care (reference=same day) | 775 (44.4%) | | | | |
| Next day | 559 (32.3%) | **1.21 (1.05, 1.4)** | **0.01** | **1.23 (1.06, 1.4)** | **0.01** |
| The second day | 223 (12.8%) | 1.01 (0.78, 1.3) | 0.94 | 1.13 (0.86, 1.4) | 0.38 |
| More than 2 days | 182 (10.5%) | 0.94 (0.73, 1.2) | 0.66 | 1.12 (0.87, 1.4) | 0.37 |
| Use an ITN (reference=does not use an ITN) | 1387 (79.8%) | 1.37 (0.99, 1.9) | 0.06 | 1.27 (0.93, 1.7) | 0.13 |
| Distance to public health facility >2 km (reference=<2 km) | 837 (48.2%) | 0.83 (0.64, 1.0) | 0.14 | 0.86 (0.67, 1.1) | 0.26 |
| Aware of a CHW in their village (reference=is not aware of a CHW in their village) | 1424 (81.9) | **1.43 (1.01, 2.0)** | **0.04** | **1.50 (1.10, 2.0)** | **0.01** |
| Patient age (reference=1–4 years) | 368 (21.1%) | | | – | – |
| 5–17 years | 752 (43.3%) | 0.76 (0.54, 1.0) | 0.12 | 0.79 (0.56, 1.1) | 0.18 |
| 18+ years | 618 (35.6%) | **0.59 (0.43, 0.8)** | **<0.01** | **0.59 (0.43, 0.8)** | **<0.01** |
| Patient is male (reference=female) | 710 (40.9%) | 1.02 (0.86, 1.2) | 0.78 | 0.94 (0.80, 1.1) | 0.43 |
| Respondent education level (reference=not completed primary) | 589 (33.9%) | | | – | – |
| Completed primary | 778 (44.8%) | 1.12 (0.93, 1.3) | 0.23 | 1.08 (0.87, 1.3) | 0.48 |
| Completed secondary | 371 (21.3%) | **1.41 (1.04, 1.9)** | **0.03** | 1.25 (0.93, 1.7) | 0.14 |
| Wealth quintile (reference=0th–20th) | 466 (26.8%) | | | – | – |
| 20th–40th | 222 (12.8%) | 1.04 (0.68, 1.6) | 0.83 | 1.02 (0.65, 1.5) | 0.94 |
| 40th–60th | 365 (21.0%) | 1.14 (0.86, 1.5) | 0.36 | 1.08 (0.82, 1.4) | 0.59 |
| 60th–80th | 355 (20.4%) | 1.10 (0.79, 1.5) | 0.56 | 1.03 (0.73, 1.4) | 0.86 |
| >80th | 330 (19.0%) | **1.59 (1.15, 2.2)** | **<0.01** | **1.53 (1.14, 2.0)** | **<0.01** |

This table refers to those who reported getting a test from any source.
Adjusted and unadjusted ORs for receiving a malaria diagnostic test, either microscopy or mRDT, from any source for the most recent febrile illness.
Bold indicate results significant to p<0.05.
CHW, community health worker; ITN, insecticide-treated net; mRDT, malaria rapid diagnostic test.

at 12 or 18 months (47.2% and 52.8%, respectively) following the initiation of the intervention. Fifty-five per cent (n=948) of participants reported having a malaria test for their recent illness, while 38.4% (n=364 of 948) of those tested reported having had the test at CHW. Among all surveyed participants, 41% (n=710) were male and 59% (n=1028) were female. A total of 21.2% (n=368) of the recent fevers were aged <5 years, 43.3% (n=752) were 5–17 years, while 36% (n=618) were >18 years. Only 21.4% (n=371) of the participants had completed their secondary education, while 45% (n=778) had completed their primary education. Overall, 42.8% lived more than 2 km from their nearest public health facility.

Demand-side factors that were associated with uptake of malaria testing for their most recent illness are examined (table 1). Respondents who delayed seeking care until the day after they fell ill had higher odds of receiving a test than those who sought care on the same day, but there was no difference between those who sought care the same day and those who waited until the second day. Being aware of local CHW increased the odds of having a test from any source, facility or CHW (adjusted OR=1.50, 95% CI: 1.10 to 2.04). Respondents from the wealthiest households were 53% more likely to have a test compared with the least wealthy households (adjusted OR=1.53 95% CI: 1.14 to 2.06). Being an adult at least

**Table 2** Association between patient characteristics and having an mRDT from a CHW before taking any treatment

| Participant characteristics of those who took a test from CHW (n=948) | Univariate analysis | | Multivariate analysis | |
|---|---|---|---|---|
| | Unadjusted OR (95% CI) | P value | Adjusted OR (95% CI) | P value |
| Time to malaria test (reference=same day) | | | – | – |
| Next day | 0.96 (0.71, 1.3) | 0.80 | 0.96 (0.69, 1.3) | 0.81 |
| The second day | 0.73 (0.44, 1.2) | 0.22 | 0.68 (0.42, 1.1) | 0.12 |
| More than 2 days | 0.81 (0.49, 1.3) | 0.41 | 0.87 (0.50, 1.5) | 0.62 |
| Use an ITN (reference=does not use an ITN) | **0.57 (0.41, 0.7)** | **<0.01** | **0.77 (0.55, 1.0)** | **0.11** |
| Distance to public health facility >2 km (reference=<2 km) | 1.32 (0.80, 2.2) | 0.28 | 1.29 (0.82, 2.0) | 0.27 |
| Patient age (reference=1–4 years) | | | – | – |
| 5–17 years | **2.43 (1.49, 3.9)** | **<0.01** | **2.39 (1.43, 4.0)** | **<0.01** |
| 18+ years | 0.98 (0.67, 1.4) | 0.91 | 1.09 (0.72, 1.6) | 0.68 |
| Patient is male (reference=female) | 1.23 (0.98, 1.5) | 0.07 | 1.02 (0.82, 1.2) | 0.85 |
| Respondent education level (reference=none) | | | – | – |
| Completed primary | 0.86 (0.69, 1.0) | 0.22 | 0.96 (0.75, 1.2) | 0.74 |
| Completed secondary | **0.45 (0.29, 0.6)** | **<0.01** | **0.68 (0.44, 1.0)** | **0.10** |
| Wealth quintile (reference=0th–20th) | | | – | – |
| 20th–40th | 1.13 (0.65, 1.9) | 0.671 | 1.11 (0.63, 1.9) | 0.72 |
| 40th–60th | 0.83 (0.63, 1.0) | 0.175 | 0.79 (0.58, 1.0) | 0.11 |
| 60th–80th | **0.67 (0.47, 0.9)** | **0.028** | **0.66 (0.45, 0.9)** | **0.03** |
| >80th | **0.26 (0.13, 0.5)** | **<0.001** | **0.32 (0.17, 0.6)** | **<0.01** |

This table refers to those who reported getting a test from a CHW.
In this analysis, awareness of a CHW in the village was not included as a covariate because 100% of individuals who had a test by a CHW were also aware of the CHW in the village. Of note, 73.5% of those tested elsewhere were also aware of a CHW in their village.
Bold indicate results significant to p<0.05.
CHW, community health worker; ITN, insecticide-treated net; mRDT, malaria rapid diagnostic test.

18 years old was negatively associated with taking a test (adjusted OR=0.59, 95% CI: 0.43 to 0.82) compared with children 1–4 years. Distance to a health facility greater than 2 km (adjusted OR=0.86, 95% CI: 0.67 to 0.11) and use of insecticide-treated net (adjusted OR=1.27, 95% CI: 0.93 to 1.74) were not significantly associated with testing uptake.

The analysis further focused on those who were tested for malaria by a CHW (table 2). Out of 948 participants who reported having a test, 38.4% reported having taken a test from a CHW (364 of 948). In 76% of the cases tested by the CHW, the participant reported contacting the CHW as the first (n=277 of 364) action taken for illness, while 11% reported taking drugs at home and then contacting the CHW. It is noted that wealthier participants prefer being tested at the health facility compared with the least wealthy households (adjusted OR for testing by a CHW=0.32, 95% CI: 0.17 to 0.62), and those who completed secondary school have lower odds of taking a test by the CHW, although this comparison did not reach statistical significance in the adjusted model (adjusted OR=0.68, 95% CI: 0.44 to 1.07). On the other hand, school-aged children between 5 and 17 years were

more than twice as likely to be tested by a CHW compared with those below 5 years old (adjusted OR=2.39, 95% CI: 1.43 to 4.01).

To understand how individual perceptions about malaria testing and treatment, malaria risk, and illness severity affected uptake of testing or testing by a CHW, the analysis focused on the final survey at the 18-month survey (table 3). After adjusting for demographic characteristics, patient perceptions were differentially related to testing overall and testing specifically with a CHW. Both confidence in AL treatment (adjusted OR=2.75, 95% CI: 0.1.54 to 4.92) and confidence in the accuracy of an mRDT performed by a CHW (adjusted OR=2.43, 95% CI: 1.12 to 5.27) were strongly positively associated with testing by a CHW. Those who reported their illness as severe were more likely to be tested (adjusted OR=2.37, 95% CI: 1.58 to 3.58) but had lower odds of testing by a CHW (adjusted OR=0.44, 95% CI: 0.22 to 0.87). Confidence in an mRDT result, either positive or negative, was not significantly associated with receiving a test for a recent illness, but those who reported high confidence in a negative mRDT result had substantially higher odds of testing with a CHW than in a facility, although this did not reach statistical

**Table 3** Relationship between participant beliefs and perceptions and uptake of testing from any source or from a CHW

| Had a malaria test (N=703) | Univariate analysis | | Multivariate analysis | |
|---|---|---|---|---|
| | Unadjusted OR (95% CI) | P value | Adjusted OR (95% CI) | P value |
| Confidence in a positive mRDT result | 1.02 (0.67, 1.5) | 0.93 | 0.87 (0.53, 1.4) | 0.57 |
| Confidence in a negative mRDT result | 1.34 (0.87, 2.0) | 0.19 | 1.2 4(0.72, 2.1) | 0.43 |
| Confidence in AL treatment | 1.27 (0.94, 1.7) | 0.12 | 1.06 (0.81, 1.4) | 0.63 |
| Trust an mRDT by a CHW as much as one at a facility | 0.85 (0.55, 1.3) | 0.48 | **0.66 (0.44, 0.9)** | **0.03** |
| Illness is severe | **2.36 (1.64, 3.3)** | **<0.001** | **2.37 (1.58, 3.5)** | **<0.001** |
| Prevalence (reference=low) | | | – | – |
| Medium | 1.51 (0.89, 2.5) | 0.13 | 1.48 (0.85, 2.5) | 0.16 |
| High | 1.1 2 (0.70, 1.8) | 0.64 | 1.05 (0.63, 1.7) | 0.85 |
| Had a test with a CHW (N=428) | Univariate analysis | | Multivariate analysis | |
| | Unadjusted OR (95% CI) | P value | Adjusted OR (95% CI) | P value |
| Confidence in a positive mRDT result | 1.45 (0.89, 2.3) | 0.13 | 1.04 (0.64, 1.7) | 0.86 |
| Confidence in a negative mRDT result | 1.61 (0.99, 2.6) | 0.06 | 1.58 (0.89, 2.8) | 0.12 |
| Confidence in AL treatment | **2.67 (1.52, 4.7)** | **<0.01** | **2.75 (1.54, 4.9)** | **<0.01** |
| Trust an mRDT by a CHW as much as one at a facility | **2.91 (1.29, 6.5)** | **0.01** | **2.43 (1.12, 5.2)** | **0.03** |
| Illness is severe | 0.60 (0.35, 1.0) | 0.06 | **0.44 (0.22, 0.8)** | **0.01** |
| Prevalence (reference=low) | | | – | – |
| Medium | **2.87 (1.48, 5.5)** | **<0.01** | **2.52 (1.26, 5.0)** | **0.01** |
| High | **2.85 (1.31, 6.2)** | **0.01** | **2.70 (1.17, 6.2)** | **0.02** |
| Confidence in a positive mRDT result | 1.45 (0.89, 2.3) | 0.13 | 1.04 (0.64, 1.7) | 0.86 |

Models are adjusted for patient age, gender, household wealth quintile and the education level of the respondent in addition to the variables in the table.
Bold indicate results significant to p<0.05.
AL, artemether–lumefantrine; CHW, community health worker; mRDT, malaria rapid diagnostic test.

significance at the 95% level (adjusted OR=1.58, 95% CI: 0.89 to 2.83). Finally, those who reported a moderate perceived prevalence in their village (ie, 4–7 fevers out of 10 would be due to malaria) or high perceived prevalence (ie, 8–10 out of 10 fevers would be due to malaria) had increasingly higher odds of receiving a test from a CHW (medium prevalence adjusted OR=2.52, 95% CI: 1.26 to 5.06, high prevalence adjusted OR=2.70, 95% CI: 1.17 to 6.22).

### Diagnostic testing by CHWs

In total, 32 404 mRDTs were conducted by the CHWs over the intervention period, and 33.7% (n=10 870) were positive. Those with a positive test received a voucher for a discounted quality-assured AL. These vouchers were redeemed at a participating outlet by 93.9% of voucher recipients. There were no instances of stock out of mRDTs among our CHWs. To explore factors that may be associated with the CHWs' ability to conduct mRDTs, we examined the association between testing volume, defined as the mean number of mRDTs conducted per month, and the different CHW demographic characteristics (table 4). Of the 244 trained CHWs with complete data, 147 (60.3%) were above 40 years of age, and 72 (29.5%) were male. There were 107 (43.8%) CHWs who had not completed secondary education.

In a multivariate analysis, formal employment, previous mRDT training, number of households and mRDT positivity rate at the cluster level were associated with volume of tests done. Sex, age, being married and education level were not significantly associated with testing volume. CHWs who were formally employed in some other capacity alongside their CHW voluntary work performed on average 1.37 more tests per month (adjusted coefficient=1.37, 95% CI: 0.05 to 2.70), but those who reported being trained previously on mRDTs performed 1.4 fewer tests per month (adjusted coefficient=1.4, 95% CI: −2.44 to −0.37). CHWs serving areas with a high proportion of positive tests (proportion of mRDT positive results >25%) tested on average two more clients per month than those in lower prevalence areas (adjusted coefficient=2.14, 95% CI: 1.05 to 3.22). CHWs who were responsible for at least 50 households tested more clients (adjusted coefficient=1.73, 95% CI: 0.70 to 2.74), although this did not increase further when the number of households exceeded 100 (adjusted coefficient=1.49, 95% CI: 0.74 to 2.24).

**Table 4** Association between mean number of tests performed per month and CHW characteristics

| Mean number of tests per month | 4.60 (SD=3.07) | Univariate analysis | | Multivariate analysis | |
|---|---|---|---|---|---|
| | | N=244 | | N=244 | |
| CHW characteristics | n (sample proportion) | Coefficient (95% CI) | P value | Coefficient (95% CI) | P value |
| Male | 72 (0.295) | −0.30 (−1.28, 0.68) | 0.52 | −0.28 (−1.24, 0.68) | 0.54 |
| Age | | | | | |
| <40 years (reference) | 97 (0.397) | – | – | – | – |
| ≥40 years | 147 (0.603) | −0.21 (−1.06, 0.63) | 0.60 | −0.42 (−1.12, 0.28) | 0.23 |
| Married (reference=unmarried/divorced) | 202 (0.828) | 0.37 (−0.57, 1.30) | 0.42 | 0.22 (−0.92, 1.35) | 0.69 |
| Education | | | | | |
| Below secondary (reference) | 107 (0.438) | – | – | – | – |
| Completed secondary | 137 (0.562) | 0.45 (−0.33, 1.23) | 0.24 | 0.23 (−0.40, 0.87) | 0.44 |
| Formal employment (reference=no formal employment) | 33 (0.135) | **1.13 (0.01, 2.28)** | **0.05** | **1.37 (0.05, 2.70)** | **0.04** |
| Previously trained in mRDT | 46 (0.188) | −0.46 (−1.64, 0.72) | 0.42 | **−1.40 (−2.44, −0.37)** | **0.01** |
| Households* | | | | | |
| <50 (reference) | 134 (0.549) | – | – | – | – |
| 50–100 | 66 (0.271) | **2.14 (0.63, 3.6)** | **<0.01** | **1.73 (0.70, 2.7)** | **<0.01** |
| >100 | 44 (0.180) | 1.18 (−0.40, 2.7) | 0.13 | **1.49 (0.74, 2.2)** | **<0.01** |
| Cluster-level test positivity† | | | | | |
| Low (≤25%; reference) | 141 (0.588) | – | – | – | – |
| High (>25%) | 103 (0.422) | **2.54 (0.97, 4.1)** | **<0.01** | **2.14 (1.05, 3.2)** | **<0.01** |

Two hundred seventy-two CHWs were trained, of which 28 were missing one or more of marital status, employment status, previous RDT training or education level. The 244 with complete data are reported here.
Bold indicate results significant to p<0.05.
*Number of households for which the CHW reported being responsible.
†Percentage of mRDTs conducted by the CHWs in that cluster, which turned positive over the study period.
CHW, community health worker; mRDT, malaria rapid diagnostic test.

Univariate and multivariate analyses were conducted to test the association between testing volume and CHW perceptions about their role at the midpoint of the intervention period (table 5). High client score was significantly associated with higher testing volume, indicating that more tests were performed by CHWs who perceived that their clients trusted their tests (adjusted coefficient=1.37, 95% CI: 0.11 to 2.62). The CHWs' competency at conducting all 20 steps of the mRDT process (mRDT score), the number of different activities to which CHWs were committed, whether they named mRDT testing as their most important activity or whether they cited extrinsic motivation (money, airtime or other) as important reasons for continuing their work were all not significantly associated with average tests performed per month.

## DISCUSSION

This study adds to the evidence that it is feasible for CHWs to offer malaria testing using mRDTs in the community. With the use of CHWs, large-scale malaria testing intervention can reach the underserved population who cannot afford to visit the health facilities for care and those who seek care in the retail sector. The existence of CHWs with the ability to test for malaria using mRDT provides a great opportunity; however, various factors may affect the demand and supply of testing using mRDT by CHWs.

Here we examine the outcomes of a community-based malaria testing to understand how to improve intervention design and maximise the impact of such programmes. The analysis was organised around the target group (community members) and their uptake or demand for the diagnostic testing intervention, and the supply of testing by the CHWs responsible for delivering the intervention. We were able to separate the determinants of uptake of malaria diagnostic testing generally from the determinants of uptake of mRDT performed by a CHW. These contrasting but complementary analyses helped us identify bottlenecks to intervention coverage at the community level.

Analysis of community survey data revealed that demand for malaria diagnostic testing before treating a fever was lower among adults and is positively associated with being in the highest wealth quintile (relative

 Kirui J, et al. BMJ Open 2023;13:e070482. doi:10.1136/bmjopen-2022-070482

**Table 5** Association between mean number of tests per month and CHW perceptions about their role at the midpoint survey

| CHW characteristics | n (sample proportion) | Univariate analysis | | Multivariate analysis | |
|---|---|---|---|---|---|
| | N=70 | Coefficient (95% CI) | P value | Coefficient (95% CI) | P value |
| mRDT score* (reference=low) | – | – | – | – | – |
| High | 54 (0.771) | 0.19 (–1.25, 1.6) | 0.78 | 0.11 (–1.34, 1.5) | 0.87 |
| Number of different activities last month (reference=≤3) | – | – | – | – | – |
| >3 activities | 19 (0.271) | 0.35 (–1.56, 2.2) | 0.70 | –0.45 (–2.04, 1.1) | 0.56 |
| Most important activity is malaria testing† | 48 (0.686) | 0.88 (–0.25, 2.0) | 0.12 | 0.07 (–1.16, 1.3) | 0.90 |
| Cited extrinsic motivators as important‡ | 28 (0.400) | 0.33 (–1.52, 2.1) | 0.71 | 0.67 (–0.92, 2.2) | 0.38 |
| Client score§ (reference=low) | – | – | – | – | – |
| High | 48 (0.686) | **1.46 (0.10, 2.8)** | **0.04** | **1.37 (0.11, 2.6)** | **0.04** |
| Years of experience as a CHW (reference=≤5 years) | – | – | – | – | – |
| >5 years | 38 (0.543) | –0.35 (–2.68, 1.9) | 0.75 | –0.51 (–2.56, 1.5) | 0.61 |

Multivariate ORs are adjusted for age, gender, education level and formal employment. The analysis includes a random sample of CHWs selected for midpoint evaluation.
Bold indicate results significant to p<0.05.
*mRDT score—-CHW scored >17 correct steps on a 20-step checklist for mRDT preparation and interpretation conducted 6 months after training.
†Equals 1 if malaria testing was cited as their most important activity or 0 if they cited any other activity as most important.
‡Equals 1 if the CHW cited money, or 0 if the CHW only cited non-monetary incentives or desire for recognition as important motivators for their role as a CHW.
§Client score was derived from four questions where the CHW reported how confident clients were in the testing they provided.
CHW, community health worker; mRDT, malaria rapid diagnostic test.

to the poorest), findings which are consistent with other published studies.[14–17] We also find that being aware of the local CHW increased demand for testing from any source. This is similar to other studies that have evaluated the contribution of CHWs to population health-seeking behaviour.[16 18] In the context of our programme, this is especially important because the testing was most often initiated by the community member seeking out a CHW rather than a door-to-door campaign initiated by the CHW. Malaria testing by a CHW provides an opportunity for reducing health disparities by ensuring the poor and underserved population get access to testing and are able to make an informed choice on how to manage their illness. Our results show that community members with a lower socioeconomic score were more likely to receive a test from a CHW compared with those from the highest wealth quintiles. Malaria testing by the CHW was free and this finding therefore aligns well with the intention of the intervention. Testing by a CHW was also more common among those with high confidence in the reliability of a test done by a CHW and also among those with high confidence in AL, the treatment offered through the CHW testing programme. Taken together, these results suggest that the community awareness of and perception of the CHW are key in improving demand. In contrast, although testing was high among those who considered their illness to be very severe, these individuals were less likely to be tested by a CHW. This result is appropriate

and consistent with the goals of the programme, which included immediate referral of more severe cases.

From the perspective of CHWs delivering malaria testing, a strong association was not observed between demographic characteristics and testing volume, but we did note that CHWs with concurrent formal employment tested more patients on average than those without, which could indicate greater visibility or mobility in the community. It was also observed that the monthly testing volume increased when the number of households under a CHW's purview exceeded 50, but we could not detect a significant increase beyond that when households exceeded 100, suggesting a threshold effect whereby more households created more demand for testing but the ability to meet the demand may be undermined by time constraints after some threshold beyond 100 households. Finally, we report that testing rates were higher in clusters with high mRDT positivity rates, possibly reflecting greater awareness of malarial illness or higher demand for malaria services in areas with higher malaria burden. This could also reflect higher confidence in a test among CHWs or community members who have experience with some tests turning positive as has been seen in the healthcare setting.[19]

A deeper analysis of CHW perception of the intervention was undertaken in a smaller random sample of participating CHWs. We show that testing volume is not affected by competing CHW-related priorities, but is associated with how

they perceive their clients' confidence in testing and the likelihood of the client accepting the mRDT result and advice. This aligns well with the picture from the demand side where we observe that confidence in an mRDT performed by a CHW is strongly associated with testing with a CHW.

This study had several limitations. First, we are not able to estimate the effect of free testing on the uptake of testing. Considering the study used diagnosis-dependent subsidies to test the rational use of AL, a similar study that deploys free testing in the absence of a conditional voucher would be required to resolve this. Similarly, it could be important to compare the effect of an intervention that delivered both the test and the treatment through the CHW rather than using a voucher redeemable in the private sector. Second, although the survey data have provided valuable insight on understanding drivers to demand and supply of malaria testing, seeking provider (CHW) and patient perspectives in a qualitative framework could provide a richer understanding of the facilitating factors and barriers to scaling up such an intervention, particularly the ways in which preferences and changes in behaviour may influence uptake of the intervention. Finally, though our survey design used repeated cross-sectional sampling, it is possible that the same household and/or febrile individual was surveyed at multiple survey time points and any resulting association was not explicitly modelled in our regression analysis. Our results are based on a representative, randomly selected population sample of recent fevers. As a result, our findings could be generalised to other contexts or service delivery programmes working through CHWs in communities where care for febrile illness is provided through a heterogeneous landscape of private, public and retail venues.

## CONCLUSION

In summary, scale-up of health interventions through CHWs is feasible and effective at reaching the poorest households. However, in order to maximise the impact of community-based interventions, it is important to recognise factors that may restrict both delivery and demand for such services. It is critical to not only create awareness of CHW services but foster trust in both the CHW's ability to deliver services and the health intervention itself. Engendering confidence in the CHW's proficiency is likely to have a synergistic effect on both demand for services and commitment to delivering services.

**Author affiliations**
[1]Academic Model Providing Access to Healthcare, Eldoret, Kenya
[2]Duke Global Health Institute, Duke University, Durham, North Carolina, USA
[3]Duke Global Health Institute, Duke Glopbal Inc, Nairobi, Kenya
[4]School of Medicine, Department of Family Medicine, Moi University College of Health Sciences, Eldoret, Kenya
[5]Center for Global Health Science & Security, Georgetown University Medical Center, Washington, District of Columbia, USA
[6]School of Public Health, Departmental of Epidemiology and Medical Statistics, Moi University College of Health Sciences, Eldoret, Kenya

**Acknowledgements** This study could not have been carried out without the exceptional attention to detail and commitment of our field team that include: I Khaoya, L Marango, E Mukeli, L Nukewa, E Wamalwa and A Wekesa. We are grateful to the study participants and the community health workers as well as the local community leadership that supported this work. We would also like to acknowledge the partnership and support of our Health Management Team colleagues in each subcounty and in particular V Kiplagat, A Kyalo, S Malenya and P Musita.

**Contributors** JK was respnsible for conceptualisation, data collection, writing original draft, revision and editing of final draft. JM was responsible for data curation, analysis design, writing of original draft, revision and editing of final draft. ES was responsible for writing of original draft, revision and editing of final draft. GA, LA, EN, JN, MB were responsible for data collection, study design, review and editing of final draft. JL and DM were responsible for conceptualisation, review and editing of final draft. WO'M was responsible for conceptualisation, funding acquisition, writing of original draft, revision and editing of final draft. WO'M is the guarantor.

**Funding** This study was supported by the National Institute of Allergy and Infectious Diseases of the National Institutes of Health (USA) (grant number R01AI110478).

**Disclaimer** The funders had no role in study design, data collection and analysis, decision to publish or preparation of this manuscript. The content is solely the responsibility of the authors and does not necessarily represent the official views of the National Institute of Allergy and Infectious Diseases or the National Institutes of Health.

**Competing interests** None declared.

**Patient and public involvement** Patients and/or the public were not involved in the design, or conduct, or reporting, or dissemination plans of this research.

**Patient consent for publication** Not required.

**Ethics approval** This study involves human participants and ethical approval was granted by Moi University Institutional Research and Ethics Committee (formal approval no. 0001403) and Duke University Institutional Review Board (Pro00063384). Participants gave informed consent to participate in the study before taking part.

**Provenance and peer review** Not commissioned; externally peer reviewed.

**Data availability statement** Data are available upon reasonable request. Trial data are available at https://datadryad.org/stash/dataset/doi:10.5061%2Fdryad. 59p4111. Additional data about CHWs presented here are available upon request to the authors.

**ORCID iDs**
Joseph Kirui http://orcid.org/0000-0001-5833-7608
Wendy O'Meara http://orcid.org/0000-0001-9455-3965

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
