## [Reviewer comments · BMJ Open]

ARTICLE DETAILS

TITLE (PROVISIONAL)	Supply- and demand-side factors influencing uptake of malaria testing services in the community: lessons for scale-up from a post-hoc analysis of a cluster-randomised, community-based trial in western Kenya
AUTHORS	Kirui, Joseph; Malinga, Josephine; Sang, Edna; Ambani, George; Abel, Lucy; Nalianya, Erick; Namae, Jane; Boyce, Matthew; Laktabai, Jeremiah; Menya, Diana; O'Meara, Wendy

VERSION 1 – REVIEW

REVIEWER	Ugwu Omale Alex Ekwueme Federal University Teaching Hospital Abakaliki (AEFUTHA), Community Medicine
REVIEW RETURNED	23-Jan-2023

GENERAL COMMENTS	Comments to the Authors of the Manuscript Titled “Supply and demand-side factors influencing uptake of malaria testing services in the community; lessons for scale-up from a cluster-randomized, community-based trial in western Kenya” This paper presents a report of secondary analyses of data from a cluster-randomised trial and provides important information on demand- and supply-side determinants of the extent of uptake in the context of a community-based intervention to increase uptake of malaria diagnostic testing in rural western Kenya. My comments and queries below are intended to enhance the paper. Title Line 2: The punctuation mark after “community” should preferably be a “colon” instead of a “semi-colon”Line 2: “randomized” should be replaced with “randomised” (the British spelling) throughout the manuscript. Abstract Results Line 34: If it was 38% of those who were tested, It should be clearly stated. For instance, the statement in line 33–34 could be stated as: 55% (n=948/1738) reported having a malaria diagnostic testing for their recent illness of which 38% were tested by a CHW.Lines 35–36: The interpretation, “belonging to a wealthy household (Adj.OR=1.53, 95%CI:1.14- 2.06) were associated with higher testing uptake” does not exactly reflect the result in table 2. This statement implies that the independent factor was dichotomous
---

	(“belonging to a wealthy household” or “not belonging to a wealthy household”) but the wealth quintile was multichotomous (had five categories) as presented in table 2 and the estimate in line 35 is for comparing the 5th category with the first category. The interpretation should be appropriate, balanced and informative. For example: belonging to the wealthiest household was associated with higher testing uptake compared to belonging to the least wealthy/poorest household (Adj.OR=1.53, 95%CI:1.14- 2.06). 3. Line 36–37: It is stated that “Poorer households were more likely to receive a test from a CHW” and estimates were not added (perhaps because there is no such estimates in table 3). Such statement can be made in the conclusion and discussion sections. In line with my comments in No. 2 above, the interpretation should be stated more correctly to reflect the result presented in table 3. Example: The wealthiest households were less likely to receive a test from a CHW compared to the poorest households (Adj. OR=0.32, 95%CI:0.17-0.60) 4. Line 37: “AL” should preferably be written in full. 5. Lines 37 and 38: Why are the adjusted odds ratio values in bold font style? This is not necessary in this section. Moreover, these are not the only estimates in the abstract section that are significant. 6. The “en dash” is more appropriate than “hyphen” in indicating “a range”, in this case, the range of the confidence intervals of the odds ratios or coefficients. “hyphen” should be replaced with “en dash” where applicable in the manuscript. 7. Lines 39–42:  (i) The estimates should be “adjusted coefficients” not “adjusted odds ratios”. (ii) It is stated: “Specific CHW attributes were associated with higher testing rates...”. The phrase “higher testing rates” should be replaced with “higher monthly number of tests performed by CHWs” for clarity and consistency with the outcome section (under methods) and the title of table 5. (iii) The estimate of “1.73” in line 40 is not consistent with the “1.37” in Table 5 and line 337. (iv) Line 40–41: To correctly reflect the result in table 5 (in line with my comments in No. 2 above), the statement “those serving more than 50 households (Adj.OR = 1.73, 95%CI:0.70-2.74)” should be reframed to: “those serving 50–100 households (compared to those serving less than 50 households) (Adj.OR = 1.73, 95%CI:0.70-2.74)” 8. Line 42: “demand side” was used here and in many other places but “demand-side” was used in the title and in line 271. “supply-side” and “demand-side” (with “hyphen”) should be used consistently. 9. Lines 42–44: The statement, “On both the supply side and the demand side, confidence in a test performed by a CHW was strongly associated with the success of the intervention” is confusing and seems to be a repetition of the result in lines 37–39. For e.g, “confidence” of who? Of only household members or of both household members and CHW? Note that “confidence of household
--	--

members” is already reported in lines 37–39 (from the result in table 4). Also, “confidence of CHW” is not reported in the abstract, so, why the “supply side” part of the statement? Also, what does “the success of the intervention” mean? Is it different from the significant increase in testing by a CHW in lines 37–39? The statement should be reframed or removed.

Conclusion

1. Line 45–46: Is the phrase “effective at increasing” preferable to “effective in increasing”?
2. Line 46: “increasing access” should be replaced with “increasing uptake” to better reflect the topic and the objective.

Strengths and limitations

Lines 56–60: These seems more like findings than strengths.

Background

1. Lines 66–71:

(i) There is only one citation at the end of the paragraph. Thus, it appears as though the citation is only for the estimate in the last sentence and not for the other estimates in the previous sentences. Is it not better to place at least one citation (even if it is the same citation) for each sentence containing an estimate? Alternatively, if all the estimates are from one source, could they not be stated in one compound sentence with only one citation?

(ii) The citations should be in superscript (not bracket) and placed after the full stop (not before the full stop) as per the journal’s style.

2. Line 80: Typo: There should be no brackets around “2010”

3. Line 85: There should be a comma after “therefore”

4. Lines 85–86: The phrase “are not diagnosed before treatment” should be replaced with “receive no parasitological diagnosis before treatment” or “receive no diagnostic testing before treatment”. Remember that there is also presumptive or clinical diagnosis.

5. Line 87: the “semi-colon” after “low” should be replaced with a “colon”

6. Line 90: “diagnosis” should be replaced with “parasitological diagnosis” or “diagnostic testing”

7. Lines 95–99: This paragraph is confusing and may have to be reframed or removed. For e.g, the “55%” in line 96 and the “21%” in line 97 have no important meaning without comparison values at baseline and/or in the control arm. The statements seem to suggest that these values were stagnated at “55%” and “21%” despite ongoing interventions and hence, the need to investigate responsible factors and how to increase the reach and impact of the interventions. It could also mean that the investigators were expecting 100% intervention effectiveness (100% tested and 0% purchased ACTs without a test) and because the outcome was short of the expectation, there was need to investigate other determinants. Also, “In our previous study” in line 96 should be replaced with “In our cluster-randomised trial”.

8. Lines 100–106: It should be stated here that 12 months and 18 months follow-up data from only the intervention clusters was used for the secondary analysis. In line 104, “us” should be inserted after “help”

	Methods In some subheadings, only the first word begins with a capital letter, e.g “Description of the main trial” and “Community surveys” but in others, more than one words begin with capital letters, e,g “Midpoint CHW Survey”, “Demand and Supply”, “Outcome Measures” etc. The first approach should be adopted throughout. Description of the main trial The number of CHWs who participated in the delivery of study intervention (mRDT testing services) per cluster should be clearly stated or summarised. Community surveys Line 167: “AL” should preferably be written in full. Midpoint CHW Survey The number of CHW surveyed per cluster should be clearly stated or summarised. Demand and Supply Line 198–199: Is “testing output” the same as “supply”? Outcome Measures  1. Line 207: The statement “This analysis looks at the clusters in the intervention arms” should not come under this section. 2. Line 208: “receive” should be replaced with “received”. 3. Line 210: “those with any test” should be replaced with “those who received a diagnostic test from any source” 4. Lines 212–214: The secondary outcome appears not to be stated correctly. The phrase “to investigate the testing volume and the factors influencing this” is not an outcome but an objective and should be removed. Consider stating it as follows: The secondary outcome was the testing volume, defined as the mean monthly number of tests performed by each CHW and measured through programme data collected during monthly supervision visits by the investigators. Statistical Analysis  1. Lines 216–218: The first sentence here is not clear and appears not to be complete (why is “12” in a bracket? A typo or a citation?). If a citation, consider stating it as follows: A mixed-effects logistic regression was used to evaluate the contribution of demand- and supply-side risk factors to the descriptive estimates of the primary outcome measures reported for the intervention clusters in another article.¹² 2. Lines 218–220:  (i) By the statement “The hierarchical nature of the data informed the adoption of multi-level modelling, since we include both individual-, and CHW- level risk factors in the analysis and account for clustering at the CU-level” appears as if both community members’ data and CHW’s data were included in the same analysis. For clarity, the statistical analysis statements regarding community members and CHW should be clearly separated. As such, the statement regarding CHW should be removed and merged with lines 238–242. (ii) The phrase “account for clustering” or “account for clustering within CU” is preferable to “account for clustering at the CU-level” (iii) Stating only “mixed-effects logistic regression to account for
--	--

clustering” would have sufficed but since “hierarchical nature” and “multi-level modelling” have also been stated, the level of hierarchical models used in the analysis (e.g level two or level three) has to be stated.

3. Lines 222–224: The statement is not framed correctly because the dependent variable to be entered into the first model must consist of all participants while the one for the second model must consist of only those tested. Consider stating it as following: To evaluate the uptake of testing, two logistic models were generated: the first model was for the primary outcome (the uptake of diagnostic testing from any source) and included all the individual participants while the second model was for the second primary outcome (the uptake of diagnostic testing from a CHW) and included only those who received a diagnostic testing from any source (this is the subset of all the individual participants).

4. Line 225: The phrase “at the CU level” is not appropriate and could be misinterpreted by the reader. Why not just state it like this: For both models, we accounted for clustering and included potential individual-level covariates like..... Or to give more details: For both models, we accounted for clustering in the univariate analysis and we accounted for both clustering and potential individual-level covariates like..... in the multivariate analysis.

5. Lines 238–242:

(i) For consistency and clarity, the phrase “a linear regression model adjusted for clustering at the CU level” could preferably be replaced with “a mixed-effects linear regression model was fitted to account for clustering”

(ii) The following phrases seem not to be framed correctly: “number of households responsible” and “number with previous CHW experience”. Should it not be the following “number of households served” or “number of households for which a CHW was responsible” and “previous experience” or “previous mRDT training”?

Results

1. Lines 274–275: The interpretation, “Respondents from the wealthiest households were 53% more likely to have a test (Adj. OR=1.53 95%CI: 1.14-2.06)” is not appropriate and not balanced. A better interpretation could be like this: “Compared to those from the least wealthy/poorest households, respondents from the wealthiest households were 53% more likely to have a test (Adj. OR=1.53 95%CI: 1.14-2.06), however, those from wealthier households were not”. From the results shown, an overall test of the effect of wealth quintile is very likely to show that wealth quintile has no significant effect on the outcome. Hence, the pairwise result (of comparing the 5th quintile to the 1st quintile should be interpreted with caution.

2. Lines 287-288: Please note that since the stated estimates are for the 5th quintile compared to the 1st quintile, a more appropriate interpretation should be specific like this: It is noted that the wealthiest participants preferred being tested at the health facility compared to the poorest participants (Adj. OR for testing at a CHW=0.32, 95%CI:0.17-0.60)

3. Lines 342–344: The interpretation could be made more appropriate. Note that compared to 50– 100 households, the test

volume actually decreased when number of households exceeded 100 (if you compare their adjusted coefficients). Please consider to modify as follows: CHWs who were responsible for 50–100 households tested more clients (compared to those responsible for less than 50 households) (Adj coefficient=1.73, 95%CI: 0.70,2.74). Although more clients were tested when the number of households exceeded 100 (compared to when the number of households was less than 50) (Adj coefficient = 1.49, 95%CI: 0.74-2.24)", this was less compared to when it was 50–100 households.

4. Line 341–342: The estimate "(Adj coefficient= 2.14, 95%CI: 0.63-342 3.64)" is not consistent with the "(Adj.OR, 2.14, 95%CI:1.05, 3.22)" in table 5 and line 42. This should be corrected.

5. Lines 272–273: The interpretation, "Respondents who delayed seeking care beyond the first 24 hours had higher odds of receiving a test" is not correct considering the result in table 2. In line with my earlier comments, the variable has four categories (not two) and the interpretation should at least reflect that fact that the categories are more than two. Moreover, the meaning of "beyond the first 24 hours" is ambiguous and cannot be said to mean "next day and beyond" which is a better way of merging the second through fourth categories of the variable. This is so because "first 24 hours" is not equivalent to "same day" (the reference category) considering the fact that for instance, if the onset of illness was in the night of day 1 and care was sought the morning of the following day (day 2), the respondent would report seeking care the next day but in terms of hours, this could be equal to or less than 12 hours. Also, if illness onset was in the morning of day 1 and care was sought the night of the following day (day 2), the respondent would report seeking care the next day but in terms of hours, this could be equal to or less than 36 hours (or even 40 hours). Therefore, the interpretation should include the exact words in the result table (or their appropriate synonyms).

6. Tables 2 and 3:

(i) What is the difference between these two categories, "After two days" and "More than two days"?

(ii) What is the difference between the independent variables, "time to seek care" in table 2 and "time to malaria test" in table 3? If both are the same variable, the name should be the same to avoid misunderstanding.

7. Lines 276–277, 288–290: The variables here have more than two categories. Please modify the interpretations in line with my earlier comments.

8. Line 288–289: Please note that "those with education above secondary school" does not have the same meaning as those who "completed secondary" in table 3 and this should be corrected. Also, the statement that this category had lower odds of taking a test at the CHW should be modified to reflect that it was not significant and the non-significant multivariate estimates in table 3 should not be in bold font style.

9. Lines 334–335: The interpretation is not correct because some univariate results in table 5 are not significant: For formal employment, previous RDT training, and greater than 100 household category the confidence intervals includes the null value. Therefore, the interpretation should only refer to the multivariate analysis. Also,

the univariate estimates for formal employment in table 5 should not be in bold font style.

10. Line 337 and table 5: The estimate of “1.37” is not consistent with the “1.73” in line 40. This should be corrected.

11. What is the difference between the dependent variables, “mean number of tests performed per month” in table 5 and “testing volume” in table 6? If both are the same variable, then why is the variable framed differently for each table?

12. Table 6: For clarity, what are the reference categories for “Most important activity is malaria testing” and “Cited extrinsic motivators as important”?

13. The association between community members’ awareness of a CHW in their village and uptake of diagnostic testing from any source (facility or CHW) is reported in table 2, lines 273–274 and 34–

36. Why was the association between participants’ awareness of a CHW and uptake of mRDT from a CHW not assessed and reported in table 3? The result of such assessment is very important because the intervention/mRDT services was delivered by the CHWs. This is a big omission and should be corrected. If being aware of a local CHW is not significantly associated with mRDT uptake from a CHW, then the increase in uptake of diagnostic testing from any source (facility or CHW) could not be said to be due to participants’ awareness of a local CHW in the village but more plausibly to other extraneous factors/confounders.

Discussion

1. Lines 384–386: The statement, “Analysis of community survey data revealed that increased demand for malaria diagnostic testing before treating a fever is positively associated to education level and wealth, but lower among adults” should be modified because only the associations with wealth and age group showed some significance in the multivariate result, the association with education did not show any significance as shown in table 2. Also, the phrase “but lower among adults” is confusing, maybe a typo. Maybe “but demand was lower among adults” was intended.

2. Lines 386–399: My comments here are in line with my comments in No. 13 under the result section:

(i) Lines 386–387: It should be clearly stated that being aware of the local CHW increased demand for testing from any source (facility or CHW) to make the statement consistent with the one in lines 273–274 and so that readers will not misunderstand it to mean increased diagnostic testing with CHW (because lines 387–388 compares the finding to other studies on CHWs contribution to population health-seeking behaviour)

(ii) Lines 388–390: It is stated, “In the context of our program, this is especially important because the testing was initiated by the community member seeking out a CHW rather than a door-to-door

campaign initiated by the CHW”. This statement implies (or could be misunderstood by the readers of this paper) that increased demand for diagnostic testing from CHW was as a result of community members seeking out a CHW due to their awareness of local CHW,

	however, such evidence is not clearly reported in this paper. Rather, this paper only reported that being aware of the local CHW increased demand for testing from any source (facility or CHW). Also, it is stated in lines 137–139 that “CHWs were free to perform tests in the location(s) of their choice (e.g., in their own homes, at clients’ homes, etc.), as long as the location allowed for proper and sufficiently private performance of the procedure”. If CHW could perform mRDT at clients’ homes, then while community members could invite CHW to their homes for mRDT, CHW could also (sometimes) take the initiative to visit households to test eligible members (and this could also explain the increased uptake of mRDT irrespective of participants’ prior awareness of CHW in their village) (iii) Lines 398–399: It is stated, “Taken together, these results suggest that the community awareness of and perception of the CHW is key in improving demand”. In line with my comments above, it should be clearly stated that community awareness of and perception of the CHW is key in improving demand for malaria testing from any source. 3. As a result of the above, it is very important for the association between community members’ awareness and uptake of mRDT from a CHW to be assessed and reported in table 3. Also, knowing exactly how the awareness question was asked could provide some insight. 4. Lines 406–410: (i) In line 407–408, “but did not increase further” should be replaced with “but decreased a bit”. Remember that compared to 50–100 households, the test volume actually decreased when the number households exceeded 100 (refer to my earlier comment in No. 3 under the result section). (ii) What are the other possible explanations for the observed decrease in test volume after the number of households served exceeded 100? (iii) What does “...the ability to meet the demand is saturated...” mean? For e.g does it mean stockout of mRDT kits? Was there any documentation during the study or field experience to substantiate this? The fact that the volume of test decreased when the number of households served exceeds 100 compared to when it was 50–100 households (observed by comparing their coefficients) perhaps indicates that there was more to it than just “saturation”. Hence the need to explore other plausible explanations. 5. Lines 425–426: In the second limitation, do you mean a qualitative study on provider and patients’ perspectives? If yes, it might be clearer to indicate so.
--	--

REVIEWER	Ruth Ashton Tulane University School of Public Health and Tropical Medicine
REVIEW RETURNED	06-Feb-2023

GENERAL COMMENTS	1. Abstract, line 28. I suggest specifying in the methods that the intervention was CHWs providing and performing RDTs on individuals with symptoms of malaria. At present, this is a bit ambiguous and could be interpreted as households being provided mRDTs for future use, or RDTs being used for mass screening and treatment (on asymptomatic individuals). 2. Abstract, line 37. “Confidence in AL treatment” may need rewording. It could be interpreted as confidence in ability to get
---

	treatment, or confidence in the effectiveness of the drug itself. 3. Abstract line 43. What is the primary indicator of 'success of the intervention'? Number of RDTs performed by CHWs (or appropriately performed), or proportion of patients receiving AL who also had a positive RDT result? While you may not be able to add much detail here, rewording this will improve clarity. 4. Introduction lines 91-92. I'm curious about the extent to which the intervention (CHWs performing RDTs in the community) was able to be targeted specifically to populations which seek treatment in the retail sector. Was this just a case of implementing in communities where there is evidence of high use of the retail sector for malaria treatment? Or the intervention was not targeted to this population as such, but the primary outcomes specifically relate to those seeking treatment in the retail sector? 5. Methods lines 109-113. It may be worthwhile to mention that this CHW model differs from other settings where the CHW is permitted to both perform the RDT and provide ACT directly to the client. 6. Methods. Another sentence or two explaining in more detail the difference between CHWs and CHEWs in Kenya would be useful to understand the operational context, if communities are already be familiar with CHWs/CHEWs, and what tasks they could perform. It would also be useful to know if the CHWs recruited for the trial were CHEWs that were effectively being 'upgraded' in their skills and responsibilities, or if CHWs and CHEWs are different individuals operating in parallel in intervention clusters. This comment also applies to lines 178-180. 7. Methods, line 123. "Random sample of households with fever in the last 4 weeks" suggest rewording to "households with at least one member with fever in the last month" or "households with fever cases in the last 4 weeks". 8. Methods line 125. The primary outcome would benefit from an additional sentence of explanation. Uptake of malaria diagnostic testing is defined as participants reporting they would be willing to get a malaria test from a CHW, or the proportion of people purchasing antimalarial drugs who first got an RDT from a CHW'? 9. Methods line 126. The secondary outcome would also benefit from rewording to improve clarity. Is it the proportion of all ACTs prescribed in the study arm that were given to individuals with a positive test? Based on the survey data or pharmacy/CHW records? 10. Methods line 150. Did the CHW intervention begin at month 1 in all clusters, or was it rolled out gradually? Line 159-160 hints that it was a gradual roll-out, but it would be helpful to have this explained. 11. Methods line 167. I suggest consistently referring to either AL (if the vouchers could only be redeemed for artemether lumefantrine) or ACT (if the voucher could be redeemed for any approved ACTs) throughout the manuscript. 12. Methods line 190. Please specify who observed the RDT, presumably this was the study team, or was it a district health team supervisor or similar? 13. Methods line 190. Did the study use existing tools (e.g. OTSS checklists) to score RDT performance, or were these tools developed specifically for the study? 14. Methods line 223-224. This is not entirely clear, particularly the denominator population for model 2 (are those not receiving a test excluded?). I suggest explicitly stating the coding for the outcome for these two models. For example, in the first model, 0 = survey participants with fever who didn't receive a test from any source; 1 = survey participants with fever who received a test from any source. 15. Methods line 227. Is "the time it took to obtain the malaria test" the time they waited after falling ill before they sought a test, or the
--	--

	time it took them to travel from home to the location where they received the test? 16. Methods lines 238-242. Can you explain further why you chose to use a linear regression model for the CHW testing volume rather than a Poisson model for count data? Assuming the model is CHW-level, it is unexpected to see “number with previous CHW experience” included as a covariate since this suggests it is a cluster-level dataset, and the cluster adjustment would not be required. In your specification of the model in line 245, indicating “individual CHW i in cluster j” would help clarify things. 17. Methods. Can you mention the process for selecting a survey participant if more than one household member reported fever in the last month. Would one member be selected randomly, or was data collected on all the people in the household with reported fever. If the second is true, do you think this additional household-level clustering may need to be incorporated in your models? 18. In table 3, suggest adding to the table caption if this was among all those who got a test from any location, or among all those who had a fever in the last 4 weeks. 19. In all tables, please be consistent in the number of decimal places in each column. I recommend 2dp for ORs and CIs, but 3dp for p values. 20. Table 5. Please specify in the methods that the outcome for the linear model is mean RDTs performed each month by the specific CHW. Currently it states only that the outcome is ‘testing volume’. 21. Results line 337. Were the CHWs who were formally employed also CHEWs (and hence expected to be more experienced and/or more well known by community members)? Or employed in some other profession alongside their CHW voluntary work? 22. Results line 346. I suggest consistently referring to the baseline, 6-month, 12-month etc. surveys, rather than using midpoint survey here. 23. Discussion lines 406-410. Do you think that this saturation effect was a result of larger clusters having poorer (average) knowledge of the CHWs, or that it is related with the distance a person is willing to travel to find the CHW for a test? 24. Similar to question 5, I would be interested to see something in the discussion relating to whether demand for testing from CHWs may be modified if the CHW model allowed them to provide treatment as well, rather than only testing. Are there any other studies exploring CHW demand in a context where CHWs can both test and treat malaria? 25. References 6 & 12 are the same
--	--

VERSION 1 – AUTHOR RESPONSE

Reviewer: 1 Dr. Ugwu Omale, Alex Ekwueme Federal University Teaching Hospital Abakaliki (AEFUTHA)

Comments to the Authors of the Manuscript Titled “Supply and demand-side factors influencing uptake of malaria testing services in the community; lessons for scale-up from a cluster-randomized, community-based trial in western Kenya”

This paper presents a report of secondary analyses of data from a cluster-randomised trial and provides important information on demand- and supply-side determinants of the extent of uptake in the context of a community-based intervention to increase uptake of malaria diagnostic testing in rural western Kenya.

My comments and queries below are intended to enhance the paper.

***NOTE: All line numbers in the responses refer to the CLEAN version of the manuscript**

Title 1.

Line 2: The punctuation mark after “community” should preferably be a “colon” instead of a “semi-colon” 2. Line 2: “randomized” should be replaced with “randomised” (the British spelling) throughout the manuscript.

Response: Thank you for this comment. I have gone ahead and included a colon after the word community and replaced “randomized” with “randomised” throughout the manuscript.

Abstract

Results

Line 34: If it was 38% of those who were tested, It should be clearly stated. For instance, the statement in line 33–34 could be stated as: 55% (n=948/1738) reported having a malaria diagnostic testing for their recent illness of which 38% were tested by a CHW.

Response: Thank you for noting this. I appreciate that this makes it clearer. I have gone ahead and revised the sentence. See line 36 of the abstract section.

2. Lines 35–36: The interpretation, “belonging to a wealthy household (Adj.OR=1.53, 95%CI:1.14-2.06) were associated with higher testing uptake” does not exactly reflect the result in table 2. This statement implies that the independent factor was dichotomous (“belonging to a wealthy household” or “not belonging to a wealthy household”) but the wealth quintile was multichotomous (had five categories) as presented in table 2 and the estimate in line 35 is for comparing the 5th category with the first category. The interpretation should be appropriate, balanced and informative. For example: belonging to the wealthiest household was associated with higher testing uptake compared to belonging to the least wealthy/poorest household (Adj.OR=1.53, 95%CI:1.14-2.06).

Response: Thank you for noting this. I have incorporated the correct interpretation. See line 38-39 of the abstract section.

3. Line 36–37: It is stated that “Poorer households were more likely to receive a test from a CHW” and estimates were not added (perhaps because there is no such estimates in table 3). Such statement can be made in the conclusion and discussion sections. In line with my comments in No. 2 above, the interpretation should be stated more correctly to reflect the result presented in table 3.

Response: Thank you for noting this. The statement about poorer households being more likely to receive a test from a CHW is just the inverse of the results in Table 3 which show that wealthier households are less likely to receive a test from a CHW. In the analysis, we used the first quintile as the reference category and the ORs range from 1.13 to 0.26. If we reversed the reference category then it would range from 3.84 to 0.88. So, our results do support this statement although the estimate is not explicit in table 3. The main point here is to contrast that wealth is positively related to testing but negatively correlated with testing by a CHW. We have rephrased this in the abstract. See line 40.

Reviewer: 2

Dr. Ruth Ashton, Tulane University School of Public Health and Tropical Medicine

Comments to the Author:

1. Abstract, line 28. I suggest specifying in the methods that the intervention was CHWs providing and performing RDTs on individuals with symptoms of malaria. At present, this is a bit ambiguous and could be interpreted as households being provided mRDTs for future use, or RDTs being used for mass screening and treatment (on asymptomatic individuals).

Response: Thank you for this, I have included the statement that CHWs provided mRDT to individuals with symptoms of malaria.

2. Abstract, line 37. "Confidence in AL treatment" may need rewording. It could be interpreted as confidence in ability to get treatment, or confidence in the effectiveness of the drug itself.

Response: Thank you for this comment. I have expounded on the word confidence in AL treatment by rewriting it to "Confidence in Artemether-lumefantrine (AL) to cure malaria". This makes it clear that we are referring to confidence in the use of AL.

3. Abstract line 43. What is the primary indicator of 'success of the intervention'? Number of RDTs performed by CHWs (or appropriately performed), or proportion of patients receiving AL who also had a positive RDT result? While you may not be able to add much detail here, rewording this will improve clarity.

Response: Thank you for this comment. I have replaced the word "the success of the intervention" with "higher mRDT testing rates".

4. Introduction lines 91-92. I'm curious about the extent to which the intervention (CHWs performing RDTs in the community) was able to be targeted specifically to populations which seek treatment in the retail sector. Was this just a case of implementing in communities where there is evidence of high use of the retail sector for malaria treatment? Or the intervention was not targeted to this population as such, but the primary outcomes specifically relate to those seeking treatment in the retail sector?

Response: This intervention was implemented in communities with high use of the retail sector for treatment of febrile illness. The community that took part in this trial go to retail outlets as their first point of care when they have a fever more than 50% of the time and these outlets prescribe antimalarials without confirmatory tests. The intervention itself was delivered at the cluster level and therefore available to anyone in an intervention cluster, regardless of their normal treatment seeking. However, the intention was that the opportunity for a free test and subsidized drug would attract individuals who might otherwise go directly to a shop. I have mentioned in the background section that in our study setting nRDT is not available in pharmacies and chemist. See in line 85.

5. Methods lines 109-113. It may be worthwhile to mention that this CHW model differs from other settings where the CHW is permitted to both perform the RDT and provide ACT directly to the client.

Response: Thank you for this comment. I have included the suggested statement in lines 142-144.

"This CHW model differs from other settings implementing community case management of malaria where the CHW is permitted to both perform the mRDT and provide antimalarials directly to the client."

6. Methods. Another sentence or two explaining in more detail the difference between CHWs and CHEWs in Kenya would be useful to understand the operational context, if communities are already be familiar with CHWs/CHEWs, and what tasks they could perform. It would also be useful to know if the CHWs recruited for the trial were CHEWs that were effectively being 'upgraded' in their skills and responsibilities, or if CHWs and CHEWs are different individuals operating in parallel in intervention clusters. This comment also applies to lines 178-180.

Response: Thank you for pointing this out. I have incorporated a brief description on the difference between CHWs and CHEWs. In Kenya the two are different. In our intervention, existing teams of CHWs working under a CHEW were trained (upgraded) to deliver the intervention (Malaria Testing). See the description in lines 128-133 in the methods section.

"In Kenya, CHWs work on a volunteer basis and there are no formal education requirements. The ability to read and write are criteria for selection by the MOH. The role of CHWs as defined by the MOH is to promote good health through providing health education, basic treatments (such as first-aid), and referrals to health care facilities. The CHWs report to Community Health Extension Workers (CHEWs) who are employees of MOH and are trained community health nurses mandated to supervise the activities of CHWs in the community units."

7. Methods, line 123. "Random sample of households with fever in the last 4 weeks" suggest rewording to "households with at least one member with fever in the last month" or "households with fever cases in the last 4 weeks".

Response: Thank you for pointing out this. I have included the word "case(s)" to read, "households with fever cases in the last 4 weeks"

8. Methods line 125. The primary outcome would benefit from an additional sentence of explanation. Uptake of malaria diagnostic testing is defined as participants reporting they would be willing to get a malaria test from a CHW, or the proportion of people purchasing antimalarial drugs who first got an RDT from a CHW'?

Thank you for this comment. The section entitled 'Outcome measures' states "defined as the proportion of fevers in the previous 4 weeks that receive a diagnostic test from any source". These are the definitions for the analysis presented in this manuscript (beginning in line 213). To avoid confusion, I have removed the sentence about outcomes in the first section of the methods which referred to the main trial and are not relevant for the current analysis.

9. Methods line 126. The secondary outcome would also benefit from rewording to improve clarity. Is it the proportion of all ACTs prescribed in the study arm that were given to individuals with a positive test? Based on the survey data or pharmacy/CHW records?

Thank you for this comment. I have removed this statement to avoid confusion. See comment 8 above.

10. Methods line 150. Did the CHW intervention begin at month 1 in all clusters, or was it rolled out gradually? Line 159-160 hints that it was a gradual roll-out, but it would be helpful to have this explained.

Line 150 defines the community surveys and not the intervention. I wish to point out that, yes the intervention (malaria testing) was rolled out over one month in only the 16 community

units (intervention CUs) and was available for about 18 months continuously. However, the household was surveyed at 4-time points (6 months, 12 months, and 18 months post-baseline) in all the 32 community units (16 Intervention and 16 control). See line 262-264 in the results section.

11. Methods line 167. I suggest consistently referring to either AL (if the vouchers could only be redeemed for artemether lumefantrine) or ACT (if the voucher could be redeemed for any approved ACTs) throughout the manuscript.

Response: Thank you for this comment. ACT has been replaced with AL.

12. Methods line 190. Please specify who observed the RDT, presumably this was the study team, or was it a district health team supervisor or similar?

Response: Thank you for this comment. I have indicated that the study supervision team were involved in the observation. See line 302.

13. Methods line 190. Did the study use existing tools (e.g. OTSS checklists) to score RDT performance, or were these tools developed specifically for the study?

Response: Thank you for this comment. I have included a brief description of the tool and indicated that this was developed by the study team based on previous research studies. See line 195-197.

14. Methods line 223-224. This is not entirely clear, particularly the denominator population for model 2 (are those not receiving a test excluded?). I suggest explicitly stating the coding for the outcome for these two models. For example, in the first model, 0 = survey participants with fever who didn't receive a test from any source; 1 = survey participants with fever who received a test from any source.

Response: Yes, those not receiving a test are excluded in model 2. We have clarified these two analysis populations as you have suggested. See line 230-232

15. Methods line 227. Is "the time it took to obtain the malaria test" the time they waited after falling ill before they sought a test, or the time it took them to travel from home to the location where they received the test?

Response: Thank you for this comment. The time it took to obtain the malaria test is the time they waited after falling ill before they sought a test. I have included this description. See lines 234-235.

16. Methods lines 238-242. Can you explain further why you chose to use a linear regression model for the CHW testing volume rather than a Poisson model for count data? Assuming the model is CHW-level, it is unexpected to see "number with previous CHW experience" included as a covariate since this suggests it is a cluster-level dataset, and the cluster adjustment would not be required. In your specification of the model in line 245, indicating "individual CHW i in cluster j " would help clarify things.

Thank you for a careful read of the methods. The listed covariates were not correct and I have revised this section. These are all CHW-level covariates, except the test positivity rate per cluster which did necessitate a cluster adjustment. We chose to use tests per month averaged over the number of months of participation because the mean of counts was far different from the variance which violates the assumptions of the poisson model.

17. Methods. Can you mention the process for selecting a survey participant if more than one household member reported fever in the last month? Would one member be selected randomly, or was data collected on all the people in the household with reported fever. If the second is true, do you think this additional household-level clustering may need to be incorporated in your models?

Only one fever per household was surveyed. In the case there were two people with fever in the last four weeks within the household and both met all the eligibility criteria, one fever was selected based on alphabetical order of their given names (not family names). We have added this additional description.

18. In table 3, suggest adding to the table caption if this was among all those who got a test from any location, or among all those who had a fever in the last 4 weeks.

Thank you for this comment. I have included the captions in tables 2 and 3.

19. In all tables, please be consistent in the number of decimal places in each column. I recommend 2dp for ORs and CIs, but 3dp for p values.

Thank you for this comment. We placed three decimal points in table 3 p-values since one of the results in wealth quantiles was less than 0.01 (i.e 0.001).

20. Table 5. Please specify in the methods that the outcome for the linear model is mean RDTs performed each month by the specific CHW. Currently it states only that the outcome is 'testing volume'.

Thank you for this comment. This is corrected.

21. Results line 337. Were the CHWs who were formally employed also CHEWs (and hence expected to be more experienced and/or more well known by community members)? Or employed in some other profession alongside their CHW voluntary work?

Thank you for this comment. Yes, it refers to CHW we were employed in some other jobs alongside their CHW voluntary work. I have included this. CHWs and CHEWs are never the same people since their required qualifications are quite different. It is rare for a CHW to be employed in a health-related job unless they are employed as cleaner in a health facility. See line 351-352.

22. Results line 346. I suggest consistently referring to the baseline, 6-month, 12-month etc. surveys, rather than using midpoint survey here.

Thank you for this comment. CHW survey was carried out at the midpoint of the study and separated from the household surveys.

23. Discussion lines 406-410. Do you think that this saturation effect was a result of larger clusters

having poorer (average) knowledge of the CHWs, or that it is related with the distance a person is willing to travel to find the CHW for a test?

Thank you for this comment. It is a good question and it would be difficult to distinguish these two. We think it is related to the number of households being served by a CHW since the number of households per CHW generally varied more than the geographic area of a cluster.

24. Similar to question 5, I would be interested to see something in the discussion relating to whether demand for testing from CHWs may be modified if the CHW model allowed them to provide treatment as well, rather than only testing. Are there any other studies exploring CHW demand in a context where CHWs can both test and treat malaria?

We have added a sentence to the discussion in lines 439-441. Our intention was to give the testing information to the client who would then act on the information. We anticipated that this might also reduce unnecessary co-prescription of antibiotics which it did (see Laktabai et al in International Journal of Public Health 2022). In Kenya, CHW activities are seen as primarily for small children and pregnant women and this is reflected in the CCMM studies in Senegal where 65% of cases treated by CHWs were less than 10 years old (Ndiaye et al Malaria Journal 2013). In contrast, we were hoping to attract working adults by linking CHWs with their usual route of care.

25. References 6 & 12 are the same

Thank you. This has been corrected

VERSION 2 – REVIEW

REVIEWER	Ugwu Omale Alex Ekwueme Federal University Teaching Hospital Abakaliki (AEFUTHA), Community Medicine
REVIEW RETURNED	10-Apr-2023

GENERAL COMMENTS	Comments to the Authors of the Manuscript Titled “Supply and demand-side factors influencing uptake of malaria testing services in the community; lessons for scale-up from a cluster-randomized, community-based trial in western Kenya” The authors did not provide a full point-by-point response to my earlier comments on the original manuscript. I do not know whether they saw all my comments or not. As a result, I cannot review the current version of the manuscript. Rather, I have re-pasted below all the comments I made regarding the original manuscript. Title  Line 2: The punctuation mark after “community” should preferably be a “colon” instead of a “semi-colon” Line 2: “randomized” should be replaced with “randomised” (the British spelling) throughout the manuscript. Abstract Results  Line 34: If it was 38% of those who were tested, It should be clearly stated. For instance, the statement in line 33–34 could be stated as: 55% (n=948/1738) reported having a malaria diagnostic testing for their recent illness of which 38% were tested by a CHW. Lines 35–36: The interpretation, “belonging to a wealthy
--

	household (Adj.OR=1.53, 95%CI:1.14-2.06) were associated with higher testing uptake” does not exactly reflect the result in table 2. This statement implies that the independent factor was dichotomous (“belonging to a wealthy household” or “not belonging to a wealthy household”) but the wealth quintile was multichotomous (had five categories) as presented in table 2 and the estimate in line 35 is for comparing the 5th category with the first category. The interpretation should be appropriate, balanced and informative. For example: belonging to the wealthiest household was associated with higher testing uptake compared to belonging to the least wealthy/poorest household (Adj.OR=1.53, 95%CI:1.14-2.06). 3. Line 36–37: It is stated that “Poorer households were more likely to receive a test from a CHW” and estimates were not added (perhaps because there is no such estimates in table 3). Such statement can be made in the conclusion and discussion sections. In line with my comments in No. 2 above, the interpretation should be stated more correctly to reflect the result presented in table 3. Example: The wealthiest households were less likely to receive a test from a CHW compared to the poorest households (Adj. OR=0.32, 95%CI:0.17-0.60) 4. Line 37: “AL” should preferably be written in full. 5. Lines 37 and 38: Why are the adjusted odds ratio values in bold font style? This is not necessary in this section. Moreover, these are not the only estimates in the abstract section that are significant. 6. The “en dash” is more appropriate than “hyphen” in indicating “a range”, in this case, the range of the confidence intervals of the odds ratios or coefficients. “hyphen” should be replaced with “en dash” where applicable in the manuscript. 7. Lines 39–42: (i) The estimates should be “adjusted coefficients” not “adjusted odds ratios”. (ii) It is stated: “Specific CHW attributes were associated with higher testing rates...”. The phrase “higher testing rates” should be replaced with “higher monthly number of tests performed by CHWs” for clarity and consistency with the outcome section (under methods) and the title of table 5. (iii) The estimate of “1.73” in line 40 is not consistent with the “1.37” in Table 5 and line 337. (iv) Line 40–41: To correctly reflect the result in table 5 (in line with my comments in No. 2 above), the statement “those serving more than 50 households (Adj.OR = 1.73, 95%CI:0.70-2.74)” should be reframed to: “those serving 50–100 households (compared to those serving less than 50 households) (Adj.OR = 1.73, 95%CI:0.70-2.74)” 8. Line 42: “demand side” was used here and in many other places but “demand-side” was used in the title and in line 271. “supply-side” and “demand-side” (with “hyphen”) should be used consistently. 9. Lines 42–44: The statement, “On both the supply side and the demand side, confidence in a test performed by a CHW was strongly associated with the success of the intervention” is confusing and seems to be a repetition of the result in lines 37–39. For e.g, “confidence” of who? Of only household members or of both household members and CHW? Note that “confidence of household members” is already reported in lines 37–39 (from the result in table 4). Also, “confidence of CHW” is not reported in the abstract, so, why the “supply side” part of the statement? Also, what does “the success of the intervention” mean? Is it different from the significant increase in testing by a CHW in lines 37–39? The statement should be reframed or removed. Conclusion
--	--

	1. Line 45–46: Is the phrase “effective at increasing” preferable to “effective in increasing”? 2. Line 46: “increasing access” should be replaced with “increasing uptake” to better reflect the topic and the objective. Strengths and limitations Lines 56–60: These seems more like findings than strengths. Background 1. Lines 66–71: (i) There is only one citation at the end of the paragraph. Thus, it appears as though the citation is only for the estimate in the last sentence and not for the other estimates in the previous sentences. Is it not better to place at least one citation (even if it is the same citation) for each sentence containing an estimate? Alternatively, if all the estimates are from one source, could they not be stated in one compound sentence with only one citation? (ii) The citations should be in superscript (not bracket) and placed after the full stop (not before the full stop) as per the journal’s style. 2. Line 80: Typo: There should be no brackets around “2010” 3. Line 85: There should be a comma after “therefore” 4. Lines 85–86: The phrase “are not diagnosed before treatment” should be replaced with “receive no parasitological diagnosis before treatment” or “receive no diagnostic testing before treatment”. Remember that there is also presumptive or clinical diagnosis. 5. Line 87: the “semi-colon” after “low” should be replaced with a “colon” 6. Line 90: “diagnosis” should be replaced with “parasitological diagnosis” or “diagnostic testing” 7. Lines 95–99: This paragraph is confusing and may have to be reframed or removed. For e.g, the “55%” in line 96 and the “21%” in line 97 have no important meaning without comparison values at baseline and/or in the control arm. The statements seem to suggest that these values were stagnated at “55%” and “21%” despite ongoing interventions and hence, the need to investigate responsible factors and how to increase the reach and impact of the interventions. It could also mean that the investigators were expecting 100% intervention effectiveness (100% tested and 0% purchased ACTs without a test) and because the outcome was short of the expectation, there was need to investigate other determinants. Also, “In our previous study” in line 96 should be replaced with “In our cluster-randomised trial”. 8. Lines 100–106: It should be stated here that 12 months and 18 months follow-up data from only the intervention clusters was used for the secondary analysis. In line 104, “us” should be inserted after “help” Methods In some subheadings, only the first word begins with a capital letter, e.g “Description of the main trial” and “Community surveys” but in others, more than one words begin with capital letters, e.g “Midpoint CHW Survey”, “Demand and Supply”, “Outcome Measures” etc. The first approach should be adopted throughout. Description of the main trial The number of CHWs who participated in the delivery of study intervention (mRDT testing services) per cluster should be clearly stated or summarised. Community surveys Line 167: “AL” should preferably be written in full. Midpoint CHW Survey The number of CHW surveyed per cluster should be clearly stated or summarised. Demand and Supply
--	--

	Line 198–199: Is “testing output” the same as “supply”? Outcome Measures  1. Line 207: The statement “This analysis looks at the clusters in the intervention arms” should not come under this section. 2. Line 208: “receive” should be replaced with “received”. 3. Line 210: “those with any test” should be replaced with “those who received a diagnostic test from any source” 4. Lines 212–214: The secondary outcome appears not to be stated correctly. The phrase “to investigate the testing volume and the factors influencing this” is not an outcome but an objective and should be removed. Consider stating it as follows: The secondary outcome was the testing volume, defined as the mean monthly number of tests performed by each CHW and measured through programme data collected during monthly supervision visits by the investigators. Statistical Analysis  1. Lines 216–218: The first sentence here is not clear and appears not to be complete (why is “12” in a bracket? A typo or a citation?). If a citation, consider stating it as follows: A mixed-effects logistic regression was used to evaluate the contribution of demand- and supply-side risk factors to the descriptive estimates of the primary outcome measures reported for the intervention clusters in another article.¹² 2. Lines 218–220:  (i) By the statement “The hierarchical nature of the data informed the adoption of multi-level modelling, since we include both individual-, and CHW- level risk factors in the analysis and account for clustering at the CU-level” appears as if both community members’ data and CHW’s data were included in the same analysis. For clarity, the statistical analysis statements regarding community members and CHW should be clearly separated. As such, the statement regarding CHW should be removed and merged with lines 238–242. (ii) The phrase “account for clustering” or “account for clustering within CU” is preferable to “account for clustering at the CU-level” (iii) Stating only “mixed-effects logistic regression to account for clustering” would have sufficed but since “hierarchical nature” and “multi-level modelling” have also been stated, the level of hierarchical models used in the analysis (e.g level two or level three) has to be stated. 3. Lines 222–224: The statement is not framed correctly because the dependent variable to be entered into the first model must consist of all participants while the one for the second model must consist of only those tested. Consider stating it as following: To evaluate the uptake of testing, two logistic models were generated: the first model was for the primary outcome (the uptake of diagnostic testing from any source) and included all the individual participants while the second model was for the second primary outcome (the uptake of diagnostic testing from a CHW) and included only those who received a diagnostic testing from any source (this is the subset of all the individual participants). 4. Line 225: The phrase “at the CU level” is not appropriate and could be misinterpreted by the reader. Why not just state it like this: For both models, we accounted for clustering and included potential individual-level covariates like..... Or to give more details: For both models, we accounted for clustering in the univariate analysis and we accounted for both clustering and potential individual-level covariates like..... in the multivariate analysis.
--	--

	5. Lines 238–242: (i) For consistency and clarity, the phrase “a linear regression model adjusted for clustering at the CU level” could preferably be replaced with “a mixed-effects linear regression model was fitted to account for clustering” (ii) The following phrases seem not to be framed correctly: “number of households responsible” and “number with previous CHW experience”. Should it not be the following “number of households served” or “number of households for which a CHW was responsible” and “previous experience” or “previous mRDT training”? Results 1. Lines 274–275: The interpretation, “Respondents from the wealthiest households were 53% more likely to have a test (Adj. OR=1.53 95%CI: 1.14-2.06)” is not appropriate and not balanced. A better interpretation could be like this: “Compared to those from the least wealthy/poorest households, respondents from the wealthiest households were 53% more likely to have a test (Adj. OR=1.53 95%CI: 1.14-2.06), however, those from wealthier households were not”. From the results shown, an overall test of the effect of wealth quintile is very likely to show that wealth quintile has no significant effect on the outcome. Hence, the pairwise result (of comparing the 5th quintile to the 1st quintile should be interpreted with caution. 2. Lines 287-288: Please note that since the stated estimates are for the 5th quintile compared to the 1st quintile, a more appropriate interpretation should be specific like this: It is noted that the wealthiest participants preferred being tested at the health facility compared to the poorest participants (Adj. OR for testing at a CHW=0.32, 95%CI:0.17-0.60) 3. Lines 342–344: The interpretation could be made more appropriate. Note that compared to 50–100 households, the test volume actually decreased when number of households exceeded 100 (if you compare their adjusted coefficients). Please consider to modify as follows: CHWs who were responsible for 50–100 households tested more clients (compared to those responsible for less than 50 households) (Adj coefficient=1.73, 95%CI: 0.70,2.74). Although more clients were tested when the number of households exceeded 100 (compared to when the number of households was less than 50) (Adj coefficient = 1.49, 95%CI: 0.74-2.24)”, this was less compared to when it was 50–100 households. 4. Line 341–342: The estimate “(Adj coefficient= 2.14, 95%CI: 0.63-3.64)” is not consistent with the “(Adj coefficient, 2.14, 95%CI:1.05, 3.22)” in table 5 and line 42. This should be corrected. 5. Lines 272–273: The interpretation, “Respondents who delayed seeking care beyond the first 24 hours had higher odds of receiving a test” is not correct considering the result in table 2. In line with my earlier comments, the variable has four categories (not two) and the interpretation should at least reflect that fact that the categories are more than two. Moreover, the meaning of “beyond the first 24 hours” is ambiguous and cannot be said to mean the “next day” (or any other category) in table 2) and “within the first 24 hours” cannot be said to mean “same day” (the reference category) in table 2. Please note that if the onset of illness was in the night of day 1 and care was sought the morning of the following day (day 2), the respondent would report seeking care the next day but in terms of hours, this could be equal to or less than 12 hours. Also, if illness onset was in the morning of day 1 and care was sought the night of the following day (day 2), the respondent would report seeking care the next day but in terms of hours, this could be equal to or less than 36 hours (or even 40 hours). Therefore, the interpretation should include the
--	--

	exact words in the result table (or their appropriate synonyms). 6. Tables 2 and 3: (i) What is the difference between these two categories, “After two days” and “More than two days”? (ii) What is the difference between the independent variables, “time to seek care” in table 2 and “time to malaria test” in table 3? If both are the same variable, the name should be the same to avoid misunderstanding. 7. Lines 276–277, 288–290: The variables here have more than two categories. Please modify the interpretations in line with my earlier comments. 8. Line 288–289: Please note that “those with education above secondary school” does not have the same meaning as those who “completed secondary” in table 3 and this should be corrected. Also, the statement that this category had lower odds of taking a test at the CHW should be modified to reflect that it was not significant and the non-significant multivariate estimates in table 3 should not be in bold font style. 9. Lines 334–335: The interpretation is not correct because some univariate results in table 5 are not significant: For formal employment, previous RDT training, and greater than 100 household category the confidence intervals includes the null value. Therefore, the interpretation should only refer to the multivariate analysis. Also, the univariate estimates for formal employment in table 5 should not be in bold font style. 10. Line 337 and table 5: The estimate of “1.37” is is not consistent with the “1.73” in line 40. This should be corrected. 11. What is the difference between the dependent variables, “mean number of tests performed per month” in table 5 and “testing volume” in table 6? If both are the same variable, then why is the variable framed differently for each table? 12. Table 6: For clarity, what are the reference categories for “Most important activity is malaria testing” and “Cited extrinsic motivators as important”? 13. The association between community members’ awareness of a CHW in their village and uptake of diagnostic testing from any source (facility or CHW) is reported in table 2, lines 273–274 and 34–36. Why was the association between participants’ awareness of a CHW and uptake of mRDT from a CHW not assessed and reported in table 3? The result of such assessment is very important because the intervention/mRDT services was delivered by the CHWs. This is a big omission and should be corrected. If being aware of a local CHW is not significantly associated with mRDT uptake from a CHW, then the increase in uptake of diagnostic testing from any source (facility or CHW) could not be said to be due to participants’ awareness of a local CHW in the village but more plausibly to other extraneous factors/confounders. Discussion 1. Lines 384–386: The statement, “Analysis of community survey data revealed that increased demand for malaria diagnostic testing before treating a fever is positively associated to education level and wealth, but lower among adults” should be modified because only the associations with wealth and age group showed some significance in the multivariate result, the association with education did not show any significance as shown in table 2. Also, the phrase “but lower among adults” is confusing, maybe a typo. Maybe “but demand was lower among adults” was intended. 2. Lines 386–399: My comments here are in line with my comments in No. 13 under the result section: (i) Lines 386–387: It should be clearly stated that being aware of the
--	---

	local CHW increased demand for testing from any source (facility or CHW) to make the statement consistent with the one in lines 273–274 and so that readers will not misunderstand it to mean increased diagnostic testing with CHW (because lines 387–388 compares the finding to other studies on CHWs contribution to population health-seeking behaviour) (ii) Lines 388–390: It is stated, “In the context of our program, this is especially important because the testing was initiated by the community member seeking out a CHW rather than a door-to-door campaign initiated by the CHW”. This statement implies (or could be misunderstood by the readers of this paper) that increased demand for diagnostic testing from CHW was as a result of community members seeking out a CHW due to their awareness of local CHW, however, such evidence is not clearly reported in this paper. Rather, this paper only reported that being aware of the local CHW increased demand for testing from any source (facility or CHW). Also, it is stated in lines 137–139 that “CHWs were free to perform tests in the location(s) of their choice (e.g., in their own homes, at clients’ homes, etc.), as long as the location allowed for proper and sufficiently private performance of the procedure”. If CHW could perform mRDT at clients’ homes, then while community members could invite CHW to their homes for mRDT, CHW could also (sometimes) take the initiative to visit households to test eligible members (and this could also explain the increased uptake of mRDT irrespective of participants’ prior awareness of CHW in their village) (iii) Lines 398–399: It is stated, “Taken together, these results suggest that the community awareness of and perception of the CHW is key in improving demand”. In line with my comments above, it should be clearly stated that community awareness of and perception of the CHW is key in improving demand for malaria testing from any source. 3. As a result of the above, it is very important for the association between community members’ awareness and uptake of mRDT from a CHW to be assessed and reported in table 3. Also, knowing exactly how the awareness question was asked could provide some insight. 4. Lines 406–410: (i) In line 407–408, “but did not increase further” should be replaced with “but decreased a bit”. Remember that compared to 50–100 households, the test volume actually decreased when the number households exceeded 100 (refer to my earlier comment in No. 3 under the result section). (ii) What are the other possible explanations for the observed decrease in test volume after the number of households served exceeded 100? (iii) What does “...the ability to meet the demand is saturated...” mean? For e.g does it mean stockout of mRDT kits? Was there any documentation during the study or field experience to substantiate this? The fact that the volume of test decreased when the number of households served exceeds 100 compared to when it was 50–100 households (observed by comparing their coefficients) perhaps indicates that there was more to it than just “saturation”. Hence the need to explore other plausible explanations. 5. Lines 425–426: In the second limitation, do you mean a qualitative study on provider and patients’ perspectives? If yes, it might be clearer to indicate so.
--	---

REVIEWER	Ruth Ashton Tulane University School of Public Health and Tropical Medicine
REVIEW RETURNED	22-Mar-2023

GENERAL COMMENTS	My previous comments have all been addressed.
---

VERSION 2 – AUTHOR RESPONSE

Reviewer: 2

Dr. Ruth Ashton, Tulane University School of Public Health and Tropical Medicine

Comments to the Author:

My previous comments have all been addressed.

Comments from Reviewers

Reviewer 1: Dr. Ugwu Omale, Alex Ekwueme Federal University Teaching Hospital Abakaliki (AEFUTHA)

Comments to the Author:

Comments to the Authors of the Manuscript Titled “Supply and demand-side factors influencing uptake of malaria testing services in the community; lessons for scale-up from a cluster-randomized, community-based trial in western Kenya”

The authors did not provide a full point-by-point response to my earlier comments on the original manuscript. I do not know whether they saw all my comments or not. As a result, I cannot review the current version of the manuscript. Rather, I have re-pasted below all the comments I made regarding the original manuscript.

Response: Sorry about this. I don't know how this error occurred.

Title

- 1) Line 2: The punctuation mark after “community” should preferably be a “colon” instead of a “semi-colon”

Response: Thank you for this comment. We have gone ahead and included a colon after the word community.

- 2) Line 2: “randomized” should be replaced with “randomised” (the British spelling) throughout the manuscript.

Response: I have replaced “randomized” with “randomised” throughout the manuscript.

Abstract

Results:

- 1) Line 34: If it was 38% of those who were tested, It should be clearly stated. For instance, the statement in line 33–34 could be stated as: 55% (n=948/1738) reported having a malaria diagnostic testing for their recent illness of which 38% were tested by a CHW.

Response: Thank you for noting this. I appreciate that this makes it clearer. I have gone ahead and revised the sentence. See line 39 of the abstract section.

- 2) Lines 35–36: The interpretation, “belonging to a wealthy household (Adj.OR=1.53, 95%CI:1.14-2.06) were associated with higher testing uptake” does not exactly reflect the result in table 2. This statement implies that the independent factor was dichotomous (“belonging to a wealthy household” or “not belonging to a wealthy household”) but the wealth quintile was multichotomous (had five categories) as presented in table 2 and the estimate in line 35 is for comparing the 5th category with the first category. The interpretation should be appropriate, balanced and informative. For example: belonging to the wealthiest household was associated with higher testing uptake compared to belonging to the least wealthy/poorest household (Adj.OR=1.53, 95%CI:1.14-2.06).

Response: Thank you for noting this. I have incorporated the correct interpretation. See line 41 of the abstract section.

- 3) Line 36–37: It is stated that “Poorer households were more likely to receive a test from a CHW” and estimates were not added (perhaps because there is no such estimates in table 3). Such statement can be made in the conclusion and discussion sections. In line with my comments in No. 2 above, the interpretation should be stated more correctly to reflect the result presented in table 3. Example: The wealthiest households were less likely to receive a test from a CHW compared to the poorest households (Adj. OR=0.32, 95%CI:0.17-0.60)

Response: Thank you for noting this. The statement about poorer households being more likely to receive a test from a CHW is just the inverse of the results in Table 3 which show that wealthier households are less likely to receive a test from a CHW. In the analysis, we used the first quintile as the reference category and the ORs range from 1.13 to 0.26. If we reversed the reference category then it would range from 3.84 to 0.88. So, our results do support this statement although the estimate is not explicit in table 3. The main point here is to contrast that wealth is positively related to testing but negatively correlated with testing by a CHW. We have rephrased this in the abstract. See line 42.

- 4) Line 37: “AL” should preferably be written in full.

Response: This has been revised. See line 50

- 5) Lines 37 and 38: Why are the adjusted odds ratio values in bold font style? This is not necessary in this section. Moreover, these are not the only estimates in the abstract section that are significant.

Response: The bold style has been removed.

- 6) The “en dash” is more appropriate than “hyphen” in indicating “a range”, in this case, the range of the confidence intervals of the odds ratios or coefficients. “hyphen” should be replaced with “en dash” where applicable in the manuscript.

Response: this has been addressed.

- 7) Lines 39–42:

- i. The estimates should be “adjusted coefficients” not “adjusted odds ratios”.

Response: Analysis presented in this section only points out to adjusted odds ratio after the edits described below.

- ii. It is stated: “Specific CHW attributes were associated with higher testing rates...”. The phrase “higher testing rates” should be replaced with “higher monthly number of tests performed by CHWs” for clarity and consistency with the outcome section (under methods) and the title of table 5.

Response: This has been edited.

- iii. The estimate of “1.73” in line 40 is not consistent with the “1.37” in Table 5 and line 337.

Response: This has been removed to avoid confusion.

- iv. Line 40–41: To correctly reflect the result in table 5 (in line with my comments in No. 2 above), the statement “those serving more than 50 households (Adj.OR = 1.73, 95%CI:0.70-2.74)” should be reframed to: “those serving 50–100 households (compared to those serving less than 50 households) (Adj.OR = 1.73, 95%CI:0.70-2.74)”

Response: This has been reworded to; “serving more than 50 households (compared to <50) and serving areas with a higher test positivity”

- 8) Line 42: “demand side” was used here and in many other places but “demand-side” was used in the title and in line 271. “supply-side” and “demand-side” (with “hyphen”) should be used consistently.

Response: This has been corrected.

- 9) Lines 42–44: The statement, “On both the supply side and the demand side, confidence in a test performed by a CHW was strongly associated with the success of the intervention” is confusing and seems to be a repetition of the result in lines 37–39. For e.g, “confidence” of who? Of only household members or of both household members and CHW? Note that “confidence of household members” is already reported in lines 37–39 (from the result in table 4). Also, “confidence of CHW” is not reported in the abstract, so, why the “supply side” part of the statement? Also, what does “the success of the intervention” mean? Is it different from the significant increase in testing by a CHW in lines 37–39? The statement should be reframed or removed.

Response: This has been revised to; “On demand-side, the confidence of the respondent in a test performed by a CHW in the community was strongly associated with higher mRDT testing rates.” We have added the word; “respondent” and removed the statement “success of the intervention” and replaced it with “higher mRDT testing rates.”

Conclusion

- 1) Line 45–46: Is the phrase “effective at increasing” preferable to “effective in increasing”?

Response: this is changed to; “effective in increasing”

- 2) Line 46: “Increasing access” should be replaced with “increasing uptake” to better reflect the topic and the objective.

Response: This has been revised to “increasing uptake”

Strengths and limitations

- 1) Lines 56–60: These seem more like findings than strengths.

Response: We have highlighted the strengths and limitations associated to this analysis rather than the findings.

Background

- 1) Lines 66–71:

- i. There is only one citation at the end of the paragraph. Thus, it appears as though the citation is only for the estimate in the last sentence and not for the other estimates in the previous sentences. Is it not better to place at least one citation (even if it is the same citation) for each sentence containing an estimate?

Alternatively, if all the estimates are from one source, could they not be stated in one compound sentence with only one citation?

Response: We have placed the same citation for all the sentences.

- ii. The citations should be in superscript (not bracket) and placed after the full stop (not before the full stop) as per the journal’s style.

Response: This has been corrected

- 2) Line 80: Typo: There should be no brackets around “2010”

Response: This has been corrected.

- 3) Line 85: There should be a comma after “therefore”

Response: The comma has been included.

- 4) Lines 85–86: The phrase “are not diagnosed before treatment” should be replaced with “receive no parasitological diagnosis before treatment” or “receive no diagnostic testing before treatment”. Remember that there is also presumptive or clinical diagnosis.

Response: This phrase has been revised as advised.

- 5) Line 87: the “semi-colon” after “low” should be replaced with a “colon”

Response: A colon has been included.

- 6) Line 90: “diagnosis” should be replaced with “parasitological diagnosis” or “diagnostic testing”

Response: this has been revised to “diagnostic testing”

- 7) Lines 95–99: This paragraph is confusing and may have to be reframed or removed. For e.g, the “55%” in line 96 and the “21%” in line 97 have no important meaning without comparison values at baseline and/or in the control arm. The statements seem to suggest that these values were stagnated at “55%” and “21%” despite ongoing interventions and hence, the need to investigate responsible factors and how to increase the reach and impact of the interventions. It could also mean that the investigators were expecting 100% intervention effectiveness (100% tested and 0% purchased ACTs without a test) and because the outcome was short of the expectation, there was need to investigate other determinants. Also, “In our previous study” in line 96 should be replaced with “In our cluster-randomised trial”.

Response: The latter interpretation is correct – we anticipated higher testing rates in our intervention arm. We tried to understand in this analysis why the testing rates did not meet our expectation. We have clarified this paragraph.

- 8) Lines 100–106: It should be stated here that 12 months and 18 months follow-up data from only the intervention clusters was used for the secondary analysis. In line 104, “us” should be inserted after “help”

Response: This has been revised to include 12th-month and 18th months follow-up data and the word “us” has been included after help.

Methods

- 1) In some subheadings, only the first word begins with a capital letter, e.g “Description of the main trial” and “Community surveys” but in others, more than one words begin with capital letters, e.g “Midpoint CHW Survey”, “Demand and Supply”, “Outcome Measures” etc. The first approach should be adopted throughout.

Response: this has been corrected for all subheadings under methods.

- 2) Description of the main trial
- i. The number of CHWs who participated in the delivery of study intervention (mRDT testing services) per cluster should be clearly stated or summarised.

Response: 272 CHWs participated, a median of 18 per cluster, with a range of 10-24. This has been added.

- ii. Community surveys

- a. Line 167: “AL” should preferably be written in full.

Response: The abbreviation for AL has been defined.

- iii. Midpoint CHW Survey

- a. The number of CHW surveyed per cluster should be clearly stated or summarised.

Response: A median of 4 per CU participated as part of the 70. Range 1-10. This has been included

iv. Demand and Supply

a. Line 198–199: Is “testing output” the same as “supply”?

Response: Yes. It now reads “testing output (i.e. supply)”

v. Outcome Measures

a. Line 207: The statement “This analysis looks at the clusters in the intervention arms” should not come under this section.

Response: We have removed it here and added it to the first sentence of the methods.

b. Line 208: “receive” should be replaced with “received”.

Response: This has been corrected

c. Line 210: “those with any test” should be replaced with “those who received a diagnostic test from any source”

Response: This clarification has been added

d. Lines 212–214: The secondary outcome appears not to be stated correctly. The phrase “to investigate the testing volume and the factors influencing this” is not an outcome but an objective and should be removed. Consider stating it as follows: The secondary outcome was the testing volume, defined as the mean monthly number of tests performed by each CHW and measured through programme data collected during monthly supervision visits by the investigators.

Response: This has been corrected as suggested

vi. Statistical analysis

a. Lines 216–218: The first sentence here is not clear and appears not to be complete (why is “12” in a bracket? A typo or a citation?). If a citation, consider stating it as follows: A mixed-effects logistic regression was used to evaluate the contribution of demand- and supply-side risk factors to the descriptive estimates of the primary outcome measures reported for the intervention clusters in another article.¹²

b. Lines 218–220:

1. By the statement “The hierarchical nature of the data informed the adoption of multi-level modelling, since we include both individual-, and CHW- level risk factors in the analysis and account for clustering at the CU-level” appears as if both community members’ data and CHW’s data were included in the same analysis. For clarity, the statistical analysis statements regarding community members and CHW should be clearly separated. As such, the statement regarding CHW should be removed and merged with lines 238–242.
2. The phrase “account for clustering” or “account for clustering within CU” is preferable to “account for clustering at the CU-level”

3. Stating only “mixed-effects logistic regression to account for clustering” would have sufficed but since “hierarchical nature” and “multi-level modelling” have also been stated, the level of hierarchical models used in the analysis (e.g level two or level three) has to be stated.
4. Lines 222–224: The statement is not framed correctly because the dependent variable to be entered into the first model must consist of all participants while the one for the second model must consist of only those tested. Consider stating it as following: To evaluate the uptake of testing, two logistic models were generated: the first model was for the primary outcome (the uptake of diagnostic testing from any source) and included all the individual participants while the second model was for the second primary outcome (the uptake of diagnostic testing from a CHW) and included only those who received a diagnostic testing from any source (this is the subset of all the individual participants).
5. Line 225: The phrase “at the CU level” is not appropriate and could be misinterpreted by the reader. Why not just state it like this: For both models, we accounted for clustering and included potential individual-level covariates like..... Or to give more details: For both models, we accounted for clustering in the univariate analysis and we accounted for both clustering and potential individual-level covariates like..... in the multivariate analysis.
6. Lines 238–242:
 - a. For consistency and clarity, the phrase “a linear regression model adjusted for clustering at the CU level” could preferably be replaced with “a mixed-effects linear regression model was fitted to account for clustering”
 - b. The following phrases seem not to be framed correctly: “number of households responsible” and “number with previous CHW experience”. Should it not be the following “number of households served” or “number of households for which a CHW was responsible” and “previous experience” or “previous mRDT training”?

Response: I think we have dealt with the concerns in this section by 1) removing hierarchical modelling language in favour of mixed effects model, 2) specifying the cluster-level adjustment, 3) separating the CHW and household analyses, and 4) clarifying that the second logistic regression is a subset of the data from the first model

Results

- 1) Lines 274–275: The interpretation, “Respondents from the wealthiest households were 53% more likely to have a test (Adj. OR=1.53 95%CI: 1.14-2.06)” is not appropriate and not balanced. A better interpretation could be like this: “Compared to those from the least wealthy/poorest households, respondents from the wealthiest households were 53% more likely to have a test (Adj. OR=1.53 95%CI: 1.14-2.06), however, those from wealthier households were not”. From the results shown, an overall test of the effect of wealth quintile is very likely to show that wealth quintile has no significant effect on the outcome. Hence, the pairwise result (of comparing the 5th quintile to the 1st quintile should be interpreted with caution.

Response: We have modified this statement

- 2) Lines 287-288: Please note that since the stated estimates are for the 5th quintile compared to the 1st quintile, a more appropriate interpretation should be specific like this: It is noted that

the wealthiest participants preferred being tested at the health facility compared to the poorest participants (Adj. OR for testing at a CHW=0.32, 95%CI:0.17-0.60)

Response: We have modified this statement

- 3) Lines 342–344: The interpretation could be made more appropriate. Note that compared to 50–100 households, the test volume actually decreased when number of households exceeded 100 (if you compare their adjusted coefficients). Please consider to modify as follows: CHWs who were responsible for 50–100 households tested more clients (compared to those responsible for less than 50 households) (Adj coefficient=1.73, 95%CI: 0.70,2.74). Although more clients were tested when the number of households exceeded 100 (compared to when the number of households was less than 50) (Adj coefficient = 1.49, 95%CI: 0.74-2.24)", this was less compared to when it was 50–100 households.

Response: In our view, the original wording is more clear. The coefficient on ">100" was not larger than the coefficient on "50-100 tests" but the confidence intervals on the 50-100 category completely cover the confidence intervals in the >100 category meaning we cannot be sure whether one is higher or lower than the other. In other words, there was not a further detectable increase in tests per month for those with more than 100 households compared to those with 50-100. We interpret this more in the discussion section.

- 4) Line 341–342: The estimate "(Adj coefficient= 2.14, 95%CI: 0.63-342 3.64)" is not consistent with the "(Adj coefficient, 2.14, 95%CI:1.05, 3.22)" in table 5 and line 42. This should be corrected.

Response: Thank you for catching this typo!

- 5) Lines 272–273: The interpretation, "Respondents who delayed seeking care beyond the first 24 hours had higher odds of receiving a test" is not correct considering the result in table 2. In line with my earlier comments, the variable has four categories (not two) and the interpretation should at least reflect that fact that the categories are more than two. Moreover, the meaning of "beyond the first 24 hours" is ambiguous and cannot be said to mean the "next day" (or any other category) in table 2) and "within the first 24 hours" cannot be said to mean "same day" (the reference category) in table 2. Please note that if the onset of illness was in the night of day 1 and care was sought the morning of the following day (day 2), the respondent would report seeking care the next day but in terms of hours, this could be equal to or less than 12 hours. Also, if illness onset was in the morning of day 1 and care was sought the night of the following day (day 2), the respondent would report seeking care the next day but in terms of hours, this could be equal to or less than 36 hours (or even 40 hours). Therefore, the interpretation should include the exact words in the result table (or their appropriate synonyms).

Response: The question in the survey was – same day, next day or after two days (in other words, the second day after illness began) and more than two days. Since the question wasn't asked in terms of hours, we can't map it onto exact hours elapsed and have simply described it the way it was collected. We have edited the description in the text for accuracy and clarity around these comparisons. It now reads: *Respondents who delayed seeking care until the day after they fell ill had a higher odds of receiving a test than those who sought care on the same day, but there was no difference between those who sought care the same day and those who waited until the second day.*

- 6) Tables 2 and 3:

- i. What is the difference between these two categories, “After two days” and “More than two days”?
Response: This now reads ‘The second day’ and ‘More than two days’
- ii. What is the difference between the independent variables, “time to seek care” in table 2 and “time to malaria test” in table 3? If both are the same variable, the name should be the same to avoid misunderstanding.
Response: They are different questions, the first being the time to the first action and the second being the time until they got a test
- iii. Lines 276–277, 288–290: The variables here have more than two categories. Please modify the interpretations in line with my earlier comments.
Response: It now includes ‘compared to the least wealthy households’
- iv. Line 288–289: Please note that “those with education above secondary school” does not have the same meaning as those who “completed secondary” in table 3 and this should be corrected. Also, the statement that this category had lower odds of taking a test at the CHW should be modified to reflect that it was not significant and the non-significant multivariate estimates in table 3 should not be in bold font style.
Response: We have corrected this in the text which now reads “those who completed secondary school have lower odds of taking a test at the CHW although this comparison did not reach statistical significance in the adjusted model.” We have removed the statement about the delay to testing being correlated with testing by a CHW because this is not supported by Table 3.
- v. Lines 334–335: The interpretation is not correct because some univariate results in table 5 are not significant: For formal employment, previous RDT training, and greater than 100 household category the confidence intervals includes the null value. Therefore, the interpretation should only refer to the multivariate analysis. Also, the univariate estimates for formal employment in table 5 should not be in bold font style.
Response: There was a typo in Table 5. Formal employment, households 50-100, high prevalence in the cluster are significant in both models. Households more than 100 and previous training are only significant in the multivariable model. In any case, we have pointed only to the multivariable in this paragraph as suggested.
- vi. Line 337 and table 5: The estimate of “1.37” is not consistent with the “1.73” in line 40. This should be corrected.
Response: The table is correct and we have fixed this in the abstract
- vii. What is the difference between the dependent variables, “mean number of tests performed per month” in table 5 and “testing volume” in table 6? If both are the same variable, then why is the variable framed differently for each table?
Response: They are the same and we have corrected this to be ‘mean number of tests per month’ in both tables.
- viii. Table 6: For clarity, what are the reference categories for “Most important activity is malaria testing” and “Cited extrinsic motivators as important”?
Response: The reference is for the former is that they did not cite malaria testing as their most important work activity and the reference for the latter is that they did not cite extrinsic motivators in the reasons why they do their work as a CHW. These clarification have been added to the footnotes.
- ix. The association between community members’ awareness of a CHW in their village and uptake of diagnostic testing from any source (facility or CHW) is reported in table 2, lines 273–274 and 34–36. Why was the association between participants’ awareness of a CHW and uptake of mRDT from a CHW not assessed and reported in table 3? The result of such assessment is very important because the intervention/mRDT services was delivered by the CHWs. This is a big omission and should be corrected. If being aware of a local CHW is not significantly associated with mRDT uptake from a CHW, then the increase in uptake of diagnostic testing from any

source (facility or CHW) could not be said to be due to participants' awareness of a local CHW in the village but more plausibly to other extraneous factors/confounders. **Response: This couldn't be included in the model because it was completely colinear with the outcome. In other words, 100% of individuals who were tested by a CHW were also aware of a CHW in their village. We have added this information as a footnote to the table as well as the fact that 73.8% of those who were tested but not at the CHW were also aware of a CHW in their village**

Discussion

- 1) Lines 384–386: The statement, “Analysis of community survey data revealed that increased demand for malaria diagnostic testing before treating a fever is positively associated to education level and wealth, but lower among adults” should be modified because only the associations with wealth and age group showed some significance in the multivariate result, the association with education did not show any significance as shown in table 2. Also, the phrase “but lower among adults” is confusing, maybe a typo. Maybe “but demand was lower among adults” was intended.
Response: This has been edited to read “was lower among adults and is positively associated with wealth”
- 2) Lines 386–399: My comments here are in line with my comments in No. 13 under the result section:
 - a. Lines 386–387: It should be clearly stated that being aware of the local CHW increased demand for testing from any source (facility or CHW) to make the statement consistent with the one in lines 273–274 and so that readers will not misunderstand it to mean increased diagnostic testing with CHW (because lines 387–388 compares the finding to other studies on CHWs contribution to population health-seeking behaviour)
Response: We have added ‘from any source’
 - b. Lines 388–390: It is stated, “In the context of our program, this is especially important because the testing was initiated by the community member seeking out a CHW rather than a door-to-door campaign initiated by the CHW”. This statement implies (or could be misunderstood by the readers of this paper) that increased demand for diagnostic testing from CHW was as a result of community members seeking out a CHW due to their awareness of local CHW, however, such evidence is not clearly reported in this paper. Rather, this paper only reported that being aware of the local CHW increased demand for testing from any source (facility or CHW). Also, it is stated in lines 137–139 that “CHWs were free to perform tests in the location(s) of their choice (e.g., in their own homes, at clients' homes, etc.), as long as the location allowed for proper and sufficiently private performance of the procedure”. If CHW could perform mRDT at clients' homes, then while community members could invite CHW to their homes for mRDT, CHW could also (sometimes) take the initiative to visit households to test eligible members (and this could also explain the increased uptake of mRDT irrespective of participants' prior awareness of CHW in their village).
Response: As noted above, everyone who had a test with a CHW was aware of a CHW however this information does not clear up the question of whether the client sought out the CHW or the CHW initiated the testing through a home visit or similar activity. However, we do have some information to shed light on this question. All of the participants in the survey were those who had a fever in the last month. We asked them what they did first for their fever. Of those tested by a CHW, 76% (277/364) said their first action was to contact a CHW were also tested by a CHW. A further 11% (n=40) said they took drugs at home first and all of these reported their second action was contacting a CHW. So we do believe that the large majority of these tests were initiated by the participant rather than by the CHW. We have added these details to the results section to

support this interpretation in the discussion. We have added ‘most often’ to the discussion to indicate that testing may have been initiated by the CHW in some few instances.

- c. Lines 398–399: It is stated, “Taken together, these results suggest that the community awareness of and perception of the CHW is key in improving demand”. In line with my comments above, it should be clearly stated that community awareness of and perception of the CHW is key in improving demand for malaria testing from any source.

Response: See above.

- d. As a result of the above, it is very important for the association between community members’ awareness and uptake of mRDT from a CHW to be assessed and reported in table 3. Also, knowing exactly how the awareness question was asked could provide some insight.

Response: See response to 13 above.

- e. Lines 406–410:

- i. In line 407–408, “but did not increase further” should be replaced with “but decreased a bit”. Remember that compared to 50–100 households, the test volume actually decreased when the number households exceeded 100 (refer to my earlier comment in No. 3 under the result section).

Response: We did not test the comparison of “50-100” to “>100” directly. We can only compare the coefficients of these two relative to <50. Although the coefficient for >100 is smaller, the full range of the 95% CIs are within the 95% CIs of the “50-100” category so it seems incorrect to describe one as smaller or larger than the other given the uncertainty estimates. I do not think that ‘decreased a bit’ is accurate for the comparison between 50-100 and >100. It is probably more accurate to describe these two categories as ‘not statistically different’.

- ii. What are the other possible explanations for the observed decrease in test volume after the number of households served exceeded 100?

Response: See above. There is not a statistically measurable difference between the 50-100 and >100 categories, most likely owing to small sample size.

- iii. What does “...the ability to meet the demand is saturated...” mean? For e.g does it mean stockout of mRDT kits? Was there any documentation during the study or field experience to substantiate this? The fact that the volume of test decreased when the number of households served exceeds 100 compared to when it was 50–100 households (observed by comparing their coefficients) perhaps indicates that there was more to it than just “saturation”. Hence the need to explore other plausible explanations.

Response: There were no stockouts at all of RDTs among our CHWs as discussed in the Result section. By saturation, we intend to mean a time constraint on the part of the CHW in terms of the ability to conduct more than a certain number of tests per month on top of their livelihood activities and other CHW work.

- iv. Lines 425–426: In the second limitation, do you mean a qualitative study on provider and patients’ perspectives? If yes, it might be clearer to indicate so.

Response: Thank you, we have added mention of a qualitative study here.

VERSION 3 – REVIEW

REVIEWER	Ugwu Omale Alex Ekwueme Federal University Teaching Hospital Abakaliki (AEFUTHA), Community Medicine
-----------------	--

GENERAL COMMENTS

Comments to the Authors of the Revised Manuscript (brjopen-2022-070482.R2) Titled “Supply and demand-side factors influencing uptake of malaria testing services in the community: lessons for scale-up from a post-hoc analysis of a cluster-randomised, community-based trial in western Kenya”

Thank you for responding to my comments and queries and for revising the paper.

General comment

In perhaps too many instances where modifications were done in line with the reviewer’s comments under a particular text line, heading, or section, those modifications were not done throughout and wherever applicable in the manuscript. This action by the authors makes the reviewer to take more time to cross-check and to repeat the same comments over and over and I do not think it is fair to the reviewer.

Abstract

Line 24: Please replace “program(s)” with “programme(s)” (the British spelling) throughout in the paper.

Why the use of many headings (design, setting, participants, primary and secondary outcome measures) instead of “methods” in lines 26–35? Was it based on the Editor’s or other reviewer’s comments? I think it is better to use one heading as it was in the original paper. The use of several headings would unnecessarily increase the word count and reduce the space for other more important information.

Results

1. Line 37: 39.0% is not consistent with the 38.4% in lines 277 and 303. I do not think “38.4%” in the manuscript text should be rounded to “39.0” in the abstract. Moreover, it was 38.396... that was rounded to 38.4 in the first place.

2. Line 39–40 (and 306–307 in the main text): The use of “wealthier” and “poorer” inappropriately indicates a (significant) “linear relationship” which is not exactly supported by the result in table 2. The current interpretation suggests that the wealthier the households, the less likely it was to receive a test from a CHW (a linear relationship (which was significant)). However, in table 2, please note that while the 5th quintile (the wealthiest) was significantly less likely compared to the reference/1st quintile (the poorest), and the reported estimate in line 40 is only for this comparison, the 2nd quintile (wealthier) actually had a higher adjusted odd to receive a test from CHW compared to the 1st quintile (poorer) Adj. OR=1.11[0.63,1.96], although, this was not statistically significant (table 2). Although, the 3rd quintile (wealthier) had a lower adjusted odd compared to the 1st quintile (poorer), it was not statistically significant. From the results in table 2, although the adj. OR for the 5th quintile was lower than for the 4th quintile which was in turn lower than for the 3rd quintile, there is no indication of statistically significant differences between these three categories if they were to be compared with each other (pairwise comparisons).

As I suggested earlier, the most appropriate interpretation would be: The wealthiest households were less likely to receive a test from a CHW compared to the poorest households (Adj. OR=0.32, 95%CI:0.17-0.62). The statement in the conclusion (lines 47–48) will even become more appropriate because “poorest” was used, not “poorer”.

3. I suggested earlier that “en dash” (instead of “hyphen”) should be used to indicate “a range” throughout in the paper but this was not done, contrary to the response of the authors. If the use of “hyphen”

	is acceptable to the Journal, then I would not insist on the change. 4. Lines 43: “performing higher monthly number of tests” would be more appropriate and informative than “higher monthly number of tests” because the latter could be misinterpreted by the reader as “monthly number of tests from any source”. If there is an issue with word count, I suggest using a single heading for the write-up in lines 26–35. 5. Lines 44: “included” should be replaced with “including” or “and included” 6. Lines 43–46: If space/word count permits, including the estimates in the results would be more informative and there would be no confusion if the estimates were stated correctly and consistently. 7. Lines 45–46: Please appropriately re-state and position the statement “On demand-side, confidence of the respondent in a test performed by a CHW was strongly associated with higher mRDT testing rates” because it is not supported by the results in table 3. Kindly note the following regarding the related results in table 3: (i) “Trust an mRDT by a CHW as much as one at a facility” had a significant positive association with “having a test with a CHW” because adj. OR was 2.43 and p was 0.03. This result is correctly interpreted in the result section in lines 324–326 and in the abstract (together with other results) in lines 40–43 (Confidence in and “perceived accuracy of an mRDT (Adj.OR=2.43, 95%CI:1.12-5.27) were positively associated with testing by a CHW”). This “testing by a CHW” means receipt of an mRDT (and was the second primary outcome of the analysis) (ii) “Trust an mRDT by a CHW as much as one at a facility” had a significant negative association with “having a malaria test (having a malaria test from any source)” because adj. OR was 0.66 and p was 0.03. This result is not even interpreted in the result section (despite the fact that it was the primary outcome) and is not correctly interpreted in the abstract in lines 45–46 because the factor was associated with lower malaria testing, not higher mRDT testing as stated (the phrase “mRDT testing rates” does not mean having a malaria test (having a malaria test from any source) but is rather a repetition of “testing by a CHW” in lines 42–43. Also note that all the results from lines 36–43 are demand-side results. Therefore, the phrase “On demand-side” should be removed and the statement in lines 45–46 should be preferably placed after the other demand-side results (before the statement (“Specific CHW attributes were associated with” Which summarized the supply-side results) and appropriately stated, for example: ..... and perceived accuracy of an mRDT result (adj. OR) were positively associated with testing by a CHW. However, confidence in a test performed by a CHW was associated with lower malaria testing from any source (Adj.OR=0.66, 0.44–0.97). This result should also be interpreted in the result section and discussed in the discussion section as appropriate. Strengths and limitations 1. In lines 57–59: the strength here is not clear. Perhaps the statement is not complete. For example: to examine the strengths and shortcomings of what? I do not seem to understand the difference between the strengths in lines 51–53 and 57–59. Both actually refer to triangulation and their separation into two statements appears not to add any value. I would preferably merge both as follows: The analyses draws on the triangulation of multidimensional data from the communities (intervention recipients) and community health workers (intervention providers) and thus provide a more complete
--	--

	picture regarding the determinants of the uptake of malaria testing services in the communities 2. Lines 54–56: The statement is not clear. Who were those that might require alternative approaches? Are they clearly described in the main text? Were they compared with those that were successfully reached in the analysis? The strength in the statement should be appropriate and obvious or the statement should be deleted. I think it should be acceptable even if the strength is stated in just one bullet point. Introduction 1. Lines 65–67: The second and third sentences should be merged, for example: In 2019, an estimated 229 million cases of malaria occurred worldwide and the World Health Organization (WHO) African Region accounted for 215 million or 94% of those cases.¹ 2. Lines 97: Please replace “randomized” with “randomised” Methods Description of the main trial Line 115: Please remember to use superscript citation here and throughout in the paper Midpoint CHW survey From the description in line 190–199 and the related questions in the CHW questionnaire (items 21, 22, 36, 37), the score should not be called “client score” but “CHW perception score regarding clients trust in the mRDT” and the former should be replaced with the latter in line 193 and wherever applicable in the paper. Statistical Analysis 1. Typo in line 233: The semi-colon should be a full stop. 2. Typo in line 237: Please replace “educational” with “education” 3. Typo in line 254: Replace “education” with “educational” Results 1. Lines 266–269: I think this paragraph should be deleted. The flow of the intervention should not be included because this paper does not present the analysis of the interventional trial. It is an unnecessary repetition of what is already described in the method section. Rather, the number of clusters that participated in the community survey can be added to line 271. For example: households were approached across 16 clusters/intervention clusters, 2. Line 279: Any need for the phrase “of the recent fevers”? Were the recent fevers different from the “surveyed participants” in line 278? If no, then the denominator (surveyed participants) in line 278 should suffice. 3. Lines 291–292: “compared to being 1–4 years old” should be included in the interpretation (similar to the correct interpretation in lines 310–312). 4. Line 306–307: I refer the authors to my earlier comments concerning lines 39–40. Typo: the upper limit of the CI should be 0.62 like it is in table 2 and lines 39–40. 5. Lines 369–372: The current interpretation is more appropriate. But note that it is the effect size (in this case the coefficient) that indicates an increase or decrease in an outcome when two categories are compared. The confidence interval of the effect size then indicate the level of precision and whether the difference is statistically significant or not. This means that if the “50–100 households” is made the reference category, the effect size/coefficient will definitely be negative for the “>100 households” and irrespective of the width of the confidence interval of this negative coefficient, only two interpretations are appropriate: (1) the
--	--

	outcome significantly decreased or (2) the outcome non-significantly decreased (there is no way “increased” or “no change” can be included in the statement). Line 370: There is a typo in the confidence interval (no “en dash”) 6. Table 1, lines 288: Please add “after” to the third and fourth categories: “the second day after”, “More than two days after”. 7. Tables 2 and 3: My earlier comments regarding the use of “hours” or “days” to care-seeking applied to all related tables/texts in the manuscript and the modifications should have equally been made across board as appropriate to prevent further comments. For example, there is still “After two days” and “More than two days” category names in the current table 2 but these should have been modified like in table 1. 8. Lines 308: “have” should be replaced with “had”. This should also be done wherever applicable in the manuscript, especially in the result and method sections. 9. Table 5 title: The statement “The Multivariate odds ratios are adjusted for age, gender, education level, formal employment” is a repetition of what is already in the legend of the table and should be deleted. The statement “The analysis includes a random sample of CHWs selected for midpoint evaluation” should preferably be moved to the table legend below. 10. Tables 1–4: For each table, only the table title (the first sentence) should be at the top, all other statements should be moved to the table legend below. 11. Lines 343–347: In line with my comment no. 1 under the result section, this description of the trial intervention delivery by the CHW should be deleted and replaced with the flow of the CHW during the midpoint CHW survey and the supervisory visits vs collection of programme data as appropriate. 12. Lines 374–376: The interpretation is confusing because of the phrase “who perceived that their clients trusted their tests”. Was the analysis only among the CHWs who perceived that their clients trusted their tests? (i) Why not state as: High CHW perception score regarding clients trust in the mRDT was significantly associated with higher testing volume performed by CHWs (ii) Then “Client score” in table 5 should be replaced with “CHW perception score” and the description in the table legend below should be: 5CHW perception score was derived from four questions that measured CHW’s perceptions of how confident clients were in the testing/mRDT they provided. Discussion 1. Line 415: The word “increased” should be deleted 2. Lines 415–433: In line with my earlier comments, the fact that the relationships between wealth/socioeconomic status and diagnostic testing from any source and from a CHW were not linear should also be discussed or clarified. 3. Lines 439–441: The explanation of what was meant by “saturation” in the authors response should be added to the manuscript. 4. Generalisability was not discussed as indicated in the STROBE checklist. 5. Please update the STROBE checklist, many page numbers do not match with the exact numbers in the paper.
--	---

VERSION 3 – AUTHOR RESPONSE

Reviewer: 1

Dr. Ugwu Omale, Alex Ekwueme Federal University Teaching Hospital Abakaliki (AEFUTHA)

Comments to the Author:

Comments to the Authors of the Revised Manuscript (bmjopen-2022-070482.R2) Titled “Supply and demand-side factors influencing uptake of malaria testing services in the community: lessons for scale-up from a post-hoc analysis of a cluster-randomised, community-based trial in western Kenya”

Thank you for responding to my comments and queries and for revising the paper.

General comment

In perhaps too many instances where modifications were done in line with the reviewer’s comments under a particular text line, heading, or section, those modifications were not done throughout and wherever applicable in the manuscript. This action by the authors makes the reviewer to take more time to cross-check and to repeat the same comments over and over and I do not think it is fair to the reviewer.

Abstract

Line 24: Please replace “program(s)” with “programme(s)” (the British spelling) throughout in the paper.

Why the use of many headings (design, setting, participants, primary and secondary outcome measures) instead of “methods” in lines 26–35? Was it based on the Editor’s or other reviewer’s comments? I think it is better to use one heading as it was in the original paper. The use of several headings would unnecessarily increase the word count and reduce the space for other more important information. [NOTE FROM THE EDITORS: Please rebut this comment - this is the standard abstract format for the journal so should be maintained]

Results

1. Line 37: 39.0% is not consistent with the 38.4% in lines 277 and 303. I do not think “38.4%” in the manuscript text should be rounded to “39.0” in the abstract. Moreover, it was 38.396... that was rounded to 38.4 in the first place.

2. Line 39–40 (and 306–307 in the main text): The use of “wealthier” and “poorer” inappropriately indicates a (significant) “linear relationship” which is not exactly supported by the result in table 2. The current interpretation suggests that the wealthier the households, the less likely it was to receive a test from a CHW (a linear relationship (which was significant)). However, in table 2, please note that while the 5th quintile (the wealthiest) was significantly less likely compared to the reference/1st quintile (the poorest), and the reported estimate in line 40 is only for this comparison, the 2nd quintile (wealthier) actually had a higher adjusted odd to receive a test from CHW compared to the 1st quintile (poorer) Adj. OR=1.11[0.63,1.96], although, this was not statistically significant (table 2). Although, the 3rd quintile (wealthier) had a lower adjusted odd compared to the 1st quintile (poorer), it was not statistically significant. From the results in table 2, although the adj. OR for the 5th quintile was lower than for the 4th quintile which was in turn lower than for the 3rd quintile, there is no indication of statistically significant differences between these three categories if they were to be compared with each other (pairwise comparisons).

As I suggested earlier, the most appropriate interpretation would be: The wealthiest households were less likely to receive a test from a CHW compared to the poorest households (Adj. OR=0.32, 95%CI:0.17-0.62). The statement in the conclusion (lines 47–48) will even become more appropriate because “poorest” was used, not “poorer”.

3. I suggested earlier that “en dash” (instead of “hyphen”) should be used to indicate “a range” throughout in the paper but this was not done, contrary to the response of the authors. If the use of “hyphen” is acceptable to the Journal, then I would not insist on the change.

4. Lines 43: “performing higher monthly number of tests” would be more appropriate and informative than “higher monthly number of tests” because the latter could be misinterpreted by the reader as “monthly number of tests from any source”. If there is an issue with word count, I suggest using a single heading for the write-up in lines 26–35.

5. Lines 44: “included” should be replaced with “including” or “and included”

6. Lines 43–46: If space/word count permits, including the estimates in the results would be more informative and there would be no confusion if the estimates were stated correctly and consistently.

7. Lines 45–46: Please appropriately re-state and position the statement “On demand-side, confidence of the respondent in a test performed by a CHW was strongly associated with higher mRDT testing rates” because it is not supported by the results in table 3. Kindly note the following regarding the related results in table 3:

(i) “Trust an mRDT by a CHW as much as one at a facility” had a significant positive association with “having a test with a CHW” because adj. OR was 2.43 and p was 0.03. This result is correctly interpreted in the result section in lines 324–326 and in the abstract (together with other results) in lines 40–43 (Confidence in and “perceived accuracy of an mRDT (Adj.OR=2.43, 95%CI:1.12-5.27) were positively associated with testing by a CHW”). This “testing by a CHW” means receipt of an mRDT (and was the second primary outcome of the analysis)

(ii) “Trust an mRDT by a CHW as much as one at a facility” had a significant negative association with “having a malaria test (having a malaria test from any source)” because adj. OR was 0.66 and p was 0.03. This result is not even interpreted in the result section (despite the fact that it was the primary outcome) and is not correctly interpreted in the abstract in lines 45–46 because the factor was associated with lower malaria testing, not higher mRDT testing as stated (the phrase “mRDT testing rates” does not mean having a malaria test (having a malaria test from any source) but is rather a repetition of “testing by a CHW” in lines 42–43.

Also note that all the results from lines 36–43 are demand-side results.

Therefore, the phrase “On demand-side” should be removed and the statement in lines 45–46 should be preferably placed after the other demand-side results (before the statement (“Specific CHW attributes were associated with” Which summarized the supply-side results) and appropriately stated, for example:

..... and perceived accuracy of an mRDT result (adj. OR) were positively associated with testing by a CHW. However, confidence in a test performed by a CHW was associated with lower malaria testing from any source (Adj.OR=0.66, 0.44–0.97).

This result should also be interpreted in the result section and discussed in the discussion section as appropriate.

Response: Regarding these many suggestions for the abstract, we note that the word count for the abstract is quite limited and we are not able to include all of these changes. We have made the corrections in point 1, 4, and 5. Importantly, we have corrected the interpretation of confidence in a CHW test that was pointed out in #7 above.

Strengths and limitations

1. In lines 57–59: the strength here is not clear. Perhaps the statement is not complete. For example: to examine the strengths and shortcomings of what?

I do not seem to understand the difference between the strengths in lines 51–53 and 57–59. Both actually refer to triangulation and their separation into two statements appears not to add any value. I would preferably merge both as follows:

The analyses draws on the triangulation of multidimensional data from the communities (intervention

recipients) and community health workers (intervention providers) and thus provide a more complete picture regarding the determinants of the uptake of malaria testing services in the communities

Response: Thank you for the suggestion. We have merged these two as requested.

2. Lines 54–56: The statement is not clear. Who were those that might require alternative approaches? Are they clearly described in the main text? Were they compared with those that were successfully reached in the analysis? The strength in the statement should be appropriate and obvious or the statement should be deleted.

I think it should be acceptable even if the strength is stated in just one bullet point.

Response: We have clarified this.

Introduction

1. Lines 65–67: The second and third sentences should be merged, for example: In 2019, an estimated 229 million cases of malaria occurred worldwide and the World Health Organization (WHO) African Region accounted for 215 million or 94% of those cases.¹

Response: This has been merged

2. Lines 97: Please replace “randomized” with “randomised”

Response: We are not aware of a requirement for the British spelling and will leave this to the copy editors if such changes are required.

Methods

Description of the main trial

Line 115: Please remember to use superscript citation here and throughout in the paper

Response: corrected

Midpoint CHW survey

From the description in line 190–199 and the related questions in the CHW questionnaire (items 21, 22, 36, 37), the score should not be called “client score” but “CHW perception score regarding clients trust in the mRDT” and the former should be replaced with the latter in line 193 and wherever applicable in the paper.

Response: We have chosen not to change this as we do not feel this adds clarity. We have clearly defined this term and use it consistently throughout.

Statistical Analysis

1. Typo in line 233: The semi-colon should be a full stop.

2. Typo in line 237: Please replace “educational” with “education”

3. Typo in line 254: Replace “education” with “educational”

Response: These typos have been corrected

Results

1. Lines 266–269: I think this paragraph should be deleted. The flow of the intervention should not be included because this paper does not present the analysis of the interventional trial. It is an unnecessary repetition of what is already described in the method section.

Rather, the number of clusters that participated in the community survey can be added to line 271. For example: households were approached across 16 clusters/intervention clusters,

Response: We have chosen to keep this paragraph to orient the reader

2. Line 279: Any need for the phrase “of the recent fevers”? Were the recent fevers different from the “surveyed participants” in line 278? If no, then the denominator (surveyed participants) in line 278 should suffice.

Response: This is a matter of style. It serves to remind the reader what we are analyzing

3. Lines 291–292: “compared to being 1–4 years old” should be included in the interpretation (similar to the correct interpretation in lines 310–312).

Response: corrected

4. Line 306–307: I refer the authors to my earlier comments concerning lines 39–40. Typo: the upper limit of the CI should be 0.62 like it is in table 2 and lines 39–40.

Response: This typo has been corrected

5. Lines 369–372: The current interpretation is more appropriate. But note that it is the effect size (in this case the coefficient) that indicates an increase or decrease in an outcome when two categories are compared. The confidence interval of the effect size then indicate the level of precision and whether the difference is statistically significant or not. This means that if the “50–100 households” is made the reference category, the effect size/coefficient will definitely be negative for the “>100 households” and irrespective of the width of the confidence interval of this negative coefficient, only two interpretations are appropriate: (1) the outcome significantly decreased or (2) the outcome non-significantly decreased (there is no way “increased” or “no change” can be included in the statement).

Line 370: There is a typo in the confidence interval (no “en dash”)

Response: This typo has been corrected

6. Table 1, lines 288: Please add “after” to the third and fourth categories: “the second day after”, “More than two days after”.

Response: We have harmonized the row labels in Tables 1 and 2.

7. Tables 2 and 3: My earlier comments regarding the use of “hours” or “days” to care-seeking applied to all related tables/texts in the manuscript and the modifications should have equally been made across board as appropriate to prevent further comments. For example, there is still “After two days” and “More than two days” category names in the current table 2 but these should have been modified like in table 1.

Response: We have harmonized the row labels in Tables 1 and 2.

8. Lines 308: “have” should be replaced with “had”. This should also be done wherever applicable in the manuscript, especially in the result and method sections.

Response: We have maintained the current style

9. Table 5 title: The statement “The Multivariate odds ratios are adjusted for age, gender, education level, formal employment” is a repetition of what is already in the legend of the table and should be deleted. The statement “The analysis includes a random sample of CHWs selected for midpoint evaluation” should preferably be moved to the table legend below.

Response: We have eliminated the repetition

10. Tables 1–4: For each table, only the table title (the first sentence) should be at the top, all other statements should be moved to the table legend below.

Response: We have chosen to apply the approach of putting information that applies to the entire table at the top and information that is specific to one row as a footnote.

11. Lines 343–347: In line with my comment no. 1 under the result section, this description of the trial intervention delivery by the CHW should be deleted and replaced with the flow of the CHW during the midpoint CHW survey and the supervisory visits vs collection of programme data as appropriate.

Response: Again we have chosen to edit this slightly but maintain this important anchor point to orient the reader to the size and scope of the original trial

12. Lines 374–376: The interpretation is confusing because of the phrase “who perceived that their clients trusted their tests”. Was the analysis only among the CHWs who perceived that their clients trusted their tests?

(i) Why not state as: High CHW perception score regarding clients trust in the mRDT was significantly associated with higher testing volume performed by CHWs

(ii) Then “Client score” in table 5 should be replaced with “CHW perception score” and the description in the table legend below should be: 5CHW perception score was derived from four questions that measured CHW’s perceptions of how confident clients were in the testing/mRDT they provided.

Response: We agree it was confusing and have clarified this

Discussion

1. Line 415: The word “increased” should be deleted

Response: This has been removed

2. Lines 415–433: In line with my earlier comments, the fact that the relationships between wealth/socioeconomic status and diagnostic testing from any source and from a CHW were not linear should also be discussed or clarified.

Response: We have clarified this

3. Lines 439–441: The explanation of what was meant by “saturation” in the authors response should be added to the manuscript.

Response: This expression has been changed for clarity.

4. Generalisability was not discussed as indicated in the STROBE checklist.

Response: This has been added.

5. Please update the STROBE checklist, many page numbers do not match with the exact numbers in the paper.

Response: The updated checklist has been uploaded.